# MAXIMUM NOISE LEVEL AS THIRD OPTIMALITY CRITERION IN BLACK-BOX OPTIMIZATION PROBLEM

## ABSTRACT

This paper is devoted to the study (common in many applications) of the black-box optimization problem, where the black-box represents a gradient-free oracle $\tilde{f}_p = f(x) + \xi_p$ providing the objective function value with some stochastic noise. Assuming that the objective function is $\mu$-strongly convex, and also not just $L$-smooth, but has a higher order of smoothness ($\beta \geq 2$) we provide a novel optimization method: *Zero-Order Accelerated Batched Stochastic Gradient Descent*, whose theoretical analysis closes the question regarding the iteration complexity, *achieving optimal estimates*. Moreover, we provide a thorough analysis of the maximum noise level, and show under which condition the maximum noise level will take into account information about batch size $B$ as well as information about the smoothness order of the function $\beta$. Finally, we show the importance of considering the maximum noise level $\Delta$ as a third optimality criterion along with the standard two on the example of a numerical experiment of interest to the machine learning community, where we compare with SOTA gradient-free algorithms.

## 1 INTRODUCTION

This paper focuses on solving a standard optimization problem:

$$f^* := \min_{x \in Q \subseteq \mathbb{R}^d} f(x), \tag{1}$$

where $f : Q \to \mathbb{R}$ is function that we want to minimize, $f^*$ is the solution, which we want to find. It is known that if there are no obstacles to compute the gradient of the objective function $f$ or to compute a higher order of the derivative of the function, then optimal first- or higher-order optimizations algorithms Nesterov (2003) should be used to solve the original optimization problem equation 1. However, if computing the function gradient $\nabla f(x)$ is impossible for any reason, then perhaps the only way to solve the original problem is to use gradient-free (zero-order) optimization algorithms Conn et al. (2009); Rios & Sahinidis (2013). Among the situations in which information about the derivatives of the objective function is unavailable are the following:

a) *non-smoothness of the objective function*. This situation is probably the most widespread among theoretical works Gasnikov et al. (2022); Alashqar et al. (2023); Kornilov et al. (2024);

b) *the desire to save computational resources*, i.e., computing the gradient $\nabla f(x)$ can sometimes be much "more expensive" than computing the objective function value $f(x)$. This situation is quite popular and extremely understandable in the real world Bogolubsky et al. (2016);

c) *inaccessibility of the function gradient*. A vivid example of this situation is the problem of creating an ideal product for a particular person Lobanov et al. (2024).

Like first-order optimization algorithms, gradient-free algorithms have the following optimality criteria: $\#N$ – the number of consecutive iterations required to achieve the desired accuracy of the solution $\varepsilon$ and $\#T$ – the total number of calls (in this case) to the gradient-free oracle, where by gradient-free/derivative-free oracle we mean that we have access only to the objective function $f(x)$ with some bounded stochastic noise $\xi_p$ ($\mathbb{E}\left[\xi_p^2\right] \leq \Delta^2$). It should be noted that because the objective function is subject to noise, the gradient-free oracle plays the role of a black box. That is why there

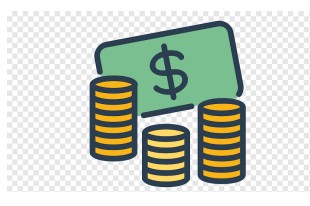
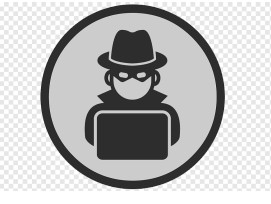
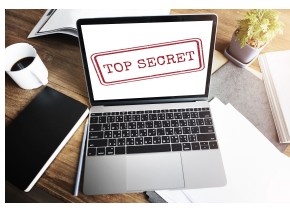

(a) Resource saving        (b) Robustness to attacks        (c) Confidentiality

Figure 1: Motivation to find the maximum noise level $\Delta$

is a tendency in the literature when the initial problem formulation equation 1 with a gradient-free oracle is called a *black-box optimization problem* Kimiaei & Neumaier (2022). However, unlike higher-order algorithms, gradient-free algorithms have a third optimality criterion: the maximum noise level $\Delta$ at which the algorithm will still converge " good", where by "good convergence" we mean convergence as in the case when $\Delta = 0$. The existence of such a seemingly unusual criterion can be explained by the following motivational examples (see Figure 1*). Among the motivations we can highlight the most demanded especially by companies (and not only). *Resource saving* (Figure 1a): The more accurately the objective function value is calculated, the more expensive this process to be performed. *Robustness to Attacks* (Figure 1b): Improving the maximum noise level makes the algorithm more robust to adversarial attacks. *Confidentiality* (Figire 1c): Some companies, due to secrecy, can't hand over all the information. Therefore, it is important to be able to answer the following question: How much can the objective function be noisy?

The basic idea to create algorithms with a gradient-free oracle that will be efficient according to the above three criteria is to take advantage of first-order algorithms by substituting a gradient approximation instead of the true gradient Gasnikov et al. (2023). The choice of the first-order optimization algorithm depends on the formulation of the original problem (on the Assumptions on the function and the gradient oracle). But the choice of gradient approximation depends on the smoothness of the function. For example, if the function is non-smooth, a smoothing scheme with $l_1$ randomization Alashqar et al. (2023); Lobanov (2023) or with $l_2$ randomization Dvinskikh et al. (2022); Lobanov et al. (2023a;b) should be used to solve the original problem. If the function is smooth, it is enough to use choose $l_1$ randomization Akhavan et al. (2022) or $l_2$ randomization Gorbunov et al. (2018); Lobanov & Gasnikov (2023). But if the objective function is not just smooth but also has a higher order of smoothness ($\beta \geq 2$), then the so-called Kernel approximation Akhavan et al. (2023); Gasnikov et al. (2024b;a), which takes into account the information about the increased smoothness of the function using two-point feedback, should be used as the gradient approximation.

In this paper, we consider the black-box optimization problem equation 1, assuming strong convexity as well as increased smoothness of the objective function. We choose accelerated stochastic gradient descent Vaswani et al. (2019) as the basis for a gradient-free algorithm. Since the Kernel approximation (which accounts for the advantages of increased smoothness) is biased, we generalize the result of Vaswani et al. (2019) to the biased gradient oracle. We use the resulting accelerated stochastic gradient descent with a biased gradient oracle to create a gradient-free algorithm. Finally, we explicitly derive estimates on the three optimality criteria of the gradient-free algorithm.

### 1.1 MAIN ASSUMPTIONS AND NOTATIONS

Since the original problem equation 1 is general, in this subsection we further define the problem by imposing constraints on the objective function as well as the zero-order oracle. In particular, we assume that the function $f$ is not just $L$-smooth, but has increased smoothness, and is also $\mu$-strongly convex.

**Assumption 1.1** (Higher order smoothness)**.** Let $l$ denote maximal integer number strictly less than $\beta$. Let $\mathcal{F}_\beta(L)$ denote the set of all functions $f : \mathbb{R}^d \to \mathbb{R}$ which are differentiable $l$ times

---

*The pictures are taken from the following resource

and $\forall x, z \in Q$ the Hölder-type condition:

$$\left| f(z) - \sum_{0 \leq |n| \leq l} \frac{1}{n!} D^n f(x)(z-x)^n \right| \leq L_\beta \|z - x\|^\beta,$$

where $l < \beta$ ($\beta$ is smoothness order), $L_\beta > 0$, the sum is over multi-index $n = (n_1, ..., n_d) \in \mathbb{N}^d$, we used the notation $n! = n_1! \cdots n_d!$, $|n| = n_1 + \cdots + n_d$, $\forall v = (v_1, ..., v_d) \in \mathbb{R}^d$, and we defined $D^n f(x) v^n = \frac{\partial^{|n|} f(x)}{\partial^{n_1} x_1 \cdots \partial^{n_d} x_d} v_1^{n_1} \cdots v_d^{n_d}$.

**Assumption 1.2** (Strongly convex). Function $f : \mathbb{R}^d \to \mathbb{R}$ is $\mu$-strongly convex with some constant $\mu > 0$ if for any $x, y \in \mathbb{R}^d$ it holds that

$$f(y) \geq f(x) + \langle \nabla f(x), y - x \rangle + \frac{\mu}{2} \|y - x\|^2.$$

Assumption 1.1 is commonly appeared in papers Bach & Perchet (2016); Akhavan et al. (2023), which consider the case when the objective function has smoothness order $\beta \geq 2$. It is worth noting that the smoothness constant $L_\beta$ in the case when $\beta = 2$ has the following relation with the standard Lipschitz gradient constant $L = 2 \cdot L_2$. In addition, Assumption 1.2 is standard among optimization works Nesterov (2003); Stich (2019).

In this paper, we assume that Algorithm 1 (which will be introduced later) only has access to the zero-order oracle, which has the following definition.

**Definition 1.3** (Zero-order oracle). The zero-order oracle $\tilde{f}_p$ returns only the objective function value $f(x_k)$ at the requested point $x_k$ with stochastic noise $\xi_p$:

$$\tilde{f}_p = f(x_k) + \xi_p,$$

where $p \in \{1, 2\}$ and we suppose that the following assumptions on stochastic noise hold

- $\xi_1 \neq \xi_2$ such that $\mathbb{E}[\xi_1^2] \leq \Delta^2$ and $\mathbb{E}\left[\xi_2^2\right] \leq \Delta^2$, where $\Delta \geq 0$ is level noise;

- the random variables $\xi_1$ and $\xi_2$ are independent from $\mathbf{e} \in S^d(1)$ is a random vector uniformly distributed on the Euclidean unit sphere, and $r$ is a random value uniformly distributed on the interval.

We impose constraints on the Kernel function which is used in Algorithm 1.

**Assumption 1.4** (Kernel function). Let function $K : [-1, 1] \to \mathbb{R}$ satisfying:

$$\mathbb{E}[K(u)] = 0, \ \mathbb{E}[uK(u)] = 1,$$
$$\mathbb{E}[u^j K(u)] = 0, \ j = 2, ..., l, \ \mathbb{E}[|u|^\beta |K(u)|] < \infty.$$

Definition 1.3 is common among gradient-free works Lobanov (2023). In particular, a zero-order oracle will produce the exact function value when the noise level is 0. We would also like to point out that we relaxed the restriction on stochastic noise by not assuming a zero mean. We only need the assumption that the random variables $\xi_1$ and $\xi_2$ are independent from $\mathbf{e}$ and $r$. Assumption 1.4 is often found in papers using the gradient approximation – the Kernel approximation. An example of such a function is the weighted sums of Lejandre polynomial Bach & Perchet (2016).

**Notation.** We use $\langle x, y \rangle := \sum_{i=1}^d x_i y_i$ to denote standard inner product of $x, y \in \mathbb{R}^d$, where $x_i$ and $y_i$ are the $i$-th component of $x$ and $y$ respectively. We denote Euclidean norm in $\mathbb{R}^d$ as $\|x\| := \sqrt{\langle x, x \rangle}$. We use the notation $B^d(r) := \left\{x \in \mathbb{R}^d : \|x\| \leq r\right\}$ to denote Euclidean ball, $S^d(r) := \left\{x \in \mathbb{R}^d : \|x\| = r\right\}$ to denote Euclidean sphere. Operator $\mathbb{E}[\cdot]$ denotes full expectation.

### 1.2 RELATED WORKS AND OUR CONTRIBUTIONS

In Table 1, we provide an overview of the convergence results of the most related works, in particular we provide estimates on the iteration complexity. Research studying the problem equation 1 with a zero-order oracle (see Definition 1.3), assuming that the function $f$ has increased smoothness

Table 1: Overview of convergence results of previous works. Notations: $d$ = dimensionality of the problem equation 1; $\beta$ = smoothness order of the objective function $f$; $\mu$ = strong convexity constant; $\varepsilon$ = desired accuracy of the problem solution by function.

| References | Iteration Complexity | Maximum Noise Level |
|---|---|---|
| Bach, Perchet (2016) Bach & Perchet (2016) | $\mathcal{O}\left(\frac{d^{2+\frac{2}{\beta-1}}\Delta^2}{\mu\varepsilon^{\frac{\beta+1}{\beta-1}}}\right)$ | ✗ |
| Akhavan, Pontil, Tsybakov (2020) Akhavan et al. (2020) | $\tilde{\mathcal{O}}\left(\frac{d^{2+\frac{2}{\beta-1}}\Delta^2}{(\mu\varepsilon)^{\frac{\beta}{\beta-1}}}\right)$ | ✗ |
| Novitskii, Gasnikov (2021) Novitskii & Gasnikov (2021) | $\tilde{\mathcal{O}}\left(\frac{d^{2+\frac{1}{\beta-1}}\Delta^2}{(\mu\varepsilon)^{\frac{\beta}{\beta-1}}}\right)$ | ✗ |
| Akhavan, Chzhen, Pontil, Tsybakov (2023) Akhavan et al. (2023) | $\tilde{\mathcal{O}}\left(\frac{d^2\Delta^2}{(\mu\varepsilon)^{\frac{\beta}{\beta-1}}}\right)$ | ✗ |
| **Theorem 3.1 (Our work)** | $\mathcal{O}\left(\sqrt{\frac{L}{\mu}}\log\frac{1}{\varepsilon}\right)$ | ✓ |

($\beta \geq 2$, see Assumption 1.1) comes from Polyak & Tsybakov (1990). After 20-30 years, this problem has received widespread attention. However, as we can see, all previous works "fought" (improved/considered) exclusively for oracle complexity (which matches the iteration complexity), without paying attention to other optimality criteria of the gradient-free algorithm. In this paper, we ask another question: Is estimation on iteration complexity unimprovable? And as we can see from Table 1 or Theorem 3.1, we significantly improve the iteration complexity without worsening the oracle complexity, and also provide the best estimates among those we have seen on $\Delta$.

More specifically, **our contributions** are the following:

- We provide a detailed explanation of the technique for creating a gradient-free algorithm that takes advantage of the increased smoothness of the function via Kernel approximation.

- We generalize existing convergence results for accelerated stochastic gradient descent to the case where the gradient oracle is biased, thereby demonstrating how bias accumulates in the convergence of the algorithm. This result may be of independent interest.

- We close the question regarding the iteration complexity search by providing an improved estimate (see Table 1) that is, we provide an optimal estimate.

- We find the maximum noise level $\Delta$ at which the algorithm will still achieve the desired accuracy $\varepsilon$ (see Table 1 and Theorem 3.1). Moreover, we show that if overbatching is done, the positive effect on the error floor is preserved in a strongly convex problem formulation.

- We show the importance of considering the maximum noise level $\Delta$ as a third optimality criterion along with the standard two using an example of a numerical experiment of interest for ML (a logistic regression problem).

**Paper Organization** This paper has the following structure. In Section 2, we present a first-order algorithm on the basis of which a novel gradient-free algorithm will be created. And in Section 3 we provide the main result of this paper, namely the convergence results of the novel accelerated gradient-free optimization algorithm. In Section 4, we provide experiments. While Section 5 concludes this paper. The missing proofs of the paper are presented in Appendix.

## 2 SEARCH FOR FIRST-ORDER ALGORITHM AS A BASE

As mentioned earlier, the basic idea of creating a gradient-free algorithm is to take advantage of first-order algorithms. That is, in this subsection, we find the first-order algorithm on which we will base to create a novel gradient-free algorithm by replacing the true gradient with a gradient approximation. Since gradient approximations use randomization on the sphere $\mathbf{e}$ (e.g., $l_1$, $l_2$ randomization, or Kernel approximation), it is important to look for a first-order algorithm that solves a stochastic optimization problem (due to the artificial stochasticity of $\mathbf{e}$). Furthermore, since the gradient approximation from a zero-order oracle concept has a bias, it is also important to find a first-order algorithm that will use a biased gradient oracle. Using these criteria, we formulate an optimization problem to find the most appropriate first-order algorithm.

## 2.1 Statement Problem

Due to the presence of artificial stochasticity in the gradient approximation, we reformulate the original optimization problem as follows:

$$f^* = \min_{x \in Q \subseteq \mathbb{R}^d} \{f(x) := \mathbb{E}\left[f(x, \xi)\right]\}. \tag{2}$$

We assume that the function satisfies the $L$-smoothness assumption, since it is a basic assumption in papers on first-order optimization algorithms.

**Assumption 2.1** ($L$-smooth). Function $f$ is $L$-smooth if it holds $\forall x, y \in \mathbb{R}^d$

$$f(y) \leq f(x) + \langle \nabla f(x), y - x \rangle + \frac{L}{2} \|y - x\|^2.$$

Next, we define a biased gradient oracle that uses a first-order algorithm.

**Definition 2.2** (Biased Gradient Oracle). A map $\mathbf{g} : \mathbb{R}^d \times \mathcal{D} \to \mathbb{R}^d$ s.t.

$$\mathbf{g}(x, \xi) = \nabla f(x, \xi) + \mathbf{b}(x)$$

for a bias $\mathbf{b} : \mathbb{R}^d \to \mathbb{R}^d$ and unbiased stochastic gradient $\mathbb{E}\left[\nabla f(x, \xi)\right] = \nabla f(x)$.

We assume that the bias and gradient noise are bounded.

**Assumption 2.3** (Bounded bias). There exists constant $\delta \geq 0$ such that $\forall x \in \mathbb{R}^d$ the following inequality holds

$$\|\mathbf{b}(x)\| = \|\mathbb{E}\left[\mathbf{g}(x, \xi)\right] - \nabla f(x)\| \leq \delta. \tag{3}$$

**Assumption 2.4** (Bounded noise). There exists constants $\rho, \sigma^2 \geq 0$ such that the more general condition of strong growth is satisfied $\forall x \in \mathbb{R}^d$

$$\mathbb{E}\left[\|\mathbf{g}(x, \xi)\|^2\right] \leq \rho \|\nabla f(x)\|^2 + \sigma^2. \tag{4}$$

Assumption 2.3 is standard for analysis, bounding bias. Assumption 2.4 is a more general condition for strong growth due to the presence of $\sigma^2$.

## 2.2 First-Order Algorithm as a Base

Now that the problem is formally defined (see Subsection 2.1), we can find an appropriate first-order algorithm. Since one of the main goals of this research is to improve the iteration complexity, we have to look for a accelerated batched first-order optimization algorithm. And the most appropriate optimization algorithm which has the following update rule:

$$x_{k+1} = y_k - \eta \mathbf{g}(y_k, \xi_k)$$
$$y_k = \alpha_k z_k + (1 - \alpha_k) x_k$$
$$z_{k+1} = \zeta_k z_k + (1 - \zeta_k) y_k - \gamma_k \eta \mathbf{g}(y_k, \xi_k).$$

has the following convergence rate presented in Lemma 2.5.

**Lemma 2.5** (Vaswani et al. (2019), Theorem 1). *Let the function $f$ satisfy Assumption 1.2 and 2.1, and the gradient oracle (see Definition 2.2 with $\delta = 0$) satisfy Assumptions 2.3 and 2.4, then with $\tilde{\rho} = \max\{1, \rho\}$ and with the chosen parameters $\gamma_k, a_{k+1}, \alpha_k, \eta$ the Accelerated Stochastic Gradient Descent has the following convergence rate:*

$$\mathbb{E}\left[f(x_N)\right] - f^* \leq \left(1 - \sqrt{\frac{\mu}{\tilde{\rho}^2 L}}\right)^N \left[f(x_0) - f^* + \frac{\mu}{2} \|x_0 - x^*\|^2\right] + \frac{\sigma^2}{\sqrt{\tilde{\rho}^2 \mu L}}.$$

As can be seen from Lemma 2.5, that the presented First Order Accelerated Algorithm is not appropriate for creating a gradient-free algorithm, since this algorithm uses an unbiased gradient oracle, and also does not use the batching technique. Therefore, we are ready to present one of the significant results of this work, namely generalizing the results of Lemma 2.5 to the case with an biased gradient oracle and also adding batching (where $B$ is a batch size).

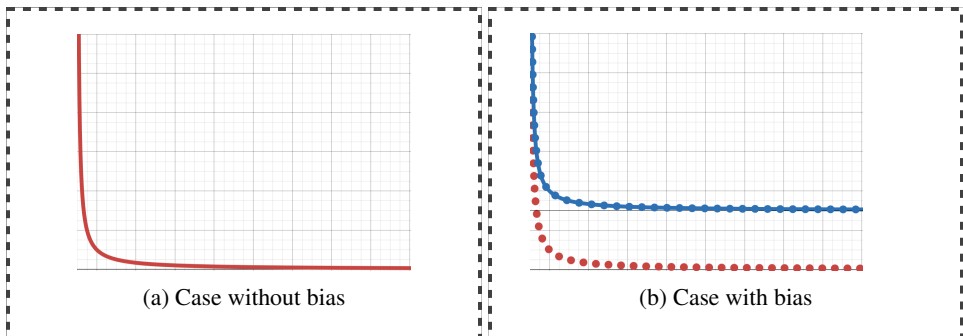

(a) Case without bias         (b) Case with bias

Figure 2: Bias influence on the algorithm convergence

**Theorem 2.6.** *Let the function $f$ satisfy Assumption 1.2 and 2.1, and the gradient oracle (see Definition 2.2) satisfy Assumptions 2.3 and 2.4, then with $\tilde{\rho}_B = \max\{1, \frac{\rho}{B}\}$ and with the chosen parameters $\gamma_k = \frac{1}{\sqrt{2\mu\eta\rho}}$, $\beta_k = 1 - \frac{\mu\eta}{2\rho}$, $b_{k+1} = \frac{\sqrt{2\mu}}{\left(1 - \sqrt{\frac{\mu\eta}{2\rho}}\right)^{(k+1)/2}}$, $a_{k+1} = \frac{1}{\left(1 - \sqrt{\frac{\mu\eta}{2\rho}}\right)^{(k+1)/2}}$, $\alpha_k = \frac{\gamma_k\beta_k b_{k+1}^2\eta}{\gamma_k\beta_k b_{k+1}^2\eta + 2a_k^2}$, $\eta \le \frac{1}{2\rho L}$ the Accelerated Stochastic Gradient Descent with batching has the following convergence rate:*

$$\mathbb{E}\left[f(x_N)\right] - f^* \le \left(1 - \sqrt{\frac{\mu}{\tilde{\rho}_B^2 L}}\right)^N \left[f(x_0) - f^* + \frac{\mu}{2}\|x_0 - x^*\|^2\right] + \frac{\sigma^2}{\sqrt{\tilde{\rho}_B^2\mu L B^2}}$$

$$+ \left(1 - \sqrt{\frac{\mu}{\tilde{\rho}_B^2 L}}\right)^N \tilde{R}\delta + \frac{\delta^2}{\sqrt{4\mu L}},$$

*where $\tilde{R} = \max_k\{\|x_k - x^*\|, \|y_k - x^*\|\}$.*

As can be seen from Theorem 2.6, this result is very similar to the result of Lemma 2.5, moreover, they will be the same if we take $\delta = 0$ and $B = 1$. It is also worth noting that the third summand does not affect convergence much (the noise does not accumulate due to the decreasing sequence), so we will not consider it in the future for simplicity. Finally, it is worth noting that the Algorithm presented in Vaswani et al. (2019) can converge as closely as possible to the problem solution (see the red line in Figure 2), while the Algorithm using the biased gradient oracle can only converge to the error floor (see the blue line in Figure 2). This is explained by the last summand from Theorem 2.6. However, convergence to the error floor opens questions about how this asymptote can be controlled. And as shown in Gasnikov et al. (2024a), the convergence of gradient-free algorithms to the asymptote depends directly on the noise level: the more noise, the better the algorithm can achieve the error floor. This fact is another clear motivation for finding the maximum noise level. For a detailed proof of Theorem 2.6, see the supplementary materials (Appendix B).

## 3 ZERO-ORDER ACCELERATED BATCHED SGD

Now that we have a proper first-order algorithm, we can move on to creating a novel gradient-free algorithm. To do this, we need to use the gradient approximation instead of the gradient oracle. In this work, we are going to use exactly the Kernel approximation because it takes into account the advantages of increased smoothness of the function, and which has the following

$$\mathbf{g}(x, \mathbf{e}) = d\frac{f(x + hr\mathbf{e}) + \xi_1 - f(x - hr\mathbf{e}) - \xi_2}{2h}K(r)\mathbf{e}, \tag{5}$$

where $h > 0$ is a smoothing parameter, $\mathbf{e} \in S^d(1)$ is a random vector uniformly distributed on the Euclidean unit sphere, $r$ is a random value uniformly distributed on the interval $r \in [0, 1]$, $K : [-1, 1] \to \mathbb{R}$ is a Kernel function. Then a novel gradient-free method aimed at solving the original problem equation 1 is presented in Algorithm 1. The missing hyperparameters are given in the Theorem 2.6.

---

**Algorithm 1** Zero-Order Accelerated Batched Stochastic Gradient Descent (ZO-ABSGD)

---

**Input:** iteration number $N$, batch size $B$, Kernel $K : [-1, 1] \to \mathbb{R}$, step size $\eta$, smoothing parameter $h$, $x_0 = y_0 = z_0 \in \mathbb{R}^d$, $a_0 = 1$, $\rho = 4d\kappa$.

    **for** $k = 0$ **to** $N - 1$ **do**

  1.  Sample vectors $\mathbf{e}_1, \mathbf{e}_2 ..., \mathbf{e}_B \in S^d(1)$ and scalars $r_1, r_2, ..., r_B \in [-1, 1]$ independently

  2.  Calculate $\mathbf{g}_k = \frac{1}{B} \sum_{i=1}^{B} \mathbf{g}(x_k, \mathbf{e}_i)$ via equation 5

  3.  $y_k \leftarrow \alpha_k z_k + (1 - \alpha_k) x_k$

  4.  $x_{k+1} \leftarrow y_k - \eta \mathbf{g}_k$

  5.  $z_{k+1} \leftarrow \beta_k z_k + (1 - \beta_k) y_k - \gamma_k \eta \mathbf{g}_k$

    **end**

**Return:** $x_N$

---

Now, in order to obtain an estimate of the convergence rate of Algorithm 1, we need to evaluate the bias as well as the second moment of the gradient approximation equation 5. Let's start with the bias of the gradient approximation:

**Bias of gradient approximation** Using the variational representation of the Euclidean norm, and definition of gradient approximation equation 5 we can write:

$$
\| \mathbb{E}\left[\mathbf{g}(x_k, \mathbf{e})\right] - \nabla f(x_k) \|
$$

$$
= \left\| \frac{d}{2h} \mathbb{E}\left[ \left( \tilde{f}(x_k + hr\mathbf{e}) - \tilde{f}(x_k - hr\mathbf{e}) \right) K(r)\mathbf{e} \right] - \nabla f(x_k) \right\|
$$

$$
\overset{①}{=} \left\| \frac{d}{h} \mathbb{E}\left[ f(x_k + hr\mathbf{e}) K(r)\mathbf{e} \right] - \nabla f(x_k) \right\|
$$

$$
\overset{②}{=} \| \mathbb{E}\left[ \nabla f(x_k + hr\mathbf{u}) r K(r) \right] - \nabla f(x_k) \|
$$

$$
= \sup_{z \in S_2^d(1)} \mathbb{E}\left[ \left( \nabla_z f(x_k + hr\mathbf{u}) - \nabla_z f(x_k) \right) r K(r) \right]
$$

$$
\overset{③,④}{\leq} \kappa_\beta h^{\beta-1} \frac{L}{(l-1)!} \mathbb{E}\left[ \|u\|^{\beta-1} \right]
$$

$$
\leq \kappa_\beta h^{\beta-1} \frac{L}{(l-1)!} \frac{d}{d + \beta - 1}
$$

$$
\lesssim \kappa_\beta L h^{\beta-1}, \tag{6}
$$

where $u \in B^d(1)$; ① = the equality is obtained from the fact, namely, distribution of $e$ is symmetric' ② = the equality is obtained from a version of Stokes' theorem Zorich & Paniagua (2016); ③ = Taylor expansion (see Appendix for more detail); ④ = assumption that $|R(hr\mathbf{u})| \leq \frac{L}{(l-1)!} \|hr\mathbf{u}\|^{\beta-1} = \frac{L}{(l-1)!} |r|^{\beta-1} h^{\beta-1} \|\mathbf{u}\|^{\beta-1}$.

Now we find an estimate of the second moment of the gradient approximation equation 5.

**Bounding second moment of gradient approximation** By definition gradient approximation equation 5 and Wirtinger-Poincare inequality we have

$$
\mathbb{E}\left[ \|\mathbf{g}(x_k, \mathbf{e})\|^2 \right] = \frac{d^2}{4h^2} \mathbb{E}\left[ \left\| \left( \tilde{f}(x_k + hr\mathbf{e}) - \tilde{f}(x_k - hr\mathbf{e}) \right) K(r)\mathbf{e} \right\|^2 \right]
$$

$$
= \frac{d^2}{4h^2} \mathbb{E}\left[ \left( f(x_k + hr\mathbf{e}) - f(x_k - hr\mathbf{e}) + (\xi_1 - \xi_2) \right)^2 K^2(r) \right]
$$

$$
\leq \frac{\kappa d^2}{2h^2} \left( \mathbb{E}\left[ \left( f(x_k + hr\mathbf{e}) - f(x_k - hr\mathbf{e}) \right)^2 \right] + 2\Delta^2 \right)
$$

$$
\leq \frac{\kappa d^2}{2h^2} \left( \frac{h^2}{d} \mathbb{E}\left[ \|\nabla f(x_k + hr\mathbf{e}) + \nabla f(x_k - hr\mathbf{e})\|^2 \right] + 2\Delta^2 \right)
$$

$$= \frac{\kappa d^2}{2h^2} \left( \frac{h^2}{d} \mathbb{E} \left[ \| \nabla f(x_k + hr\mathbf{e}) + \nabla f(x_k - hr\mathbf{e}) \pm 2\nabla f(x_k) \|^2 \right] + 2\Delta^2 \right)$$

$$\leq \underbrace{4d\kappa}_{\rho} \| \nabla f(x_k) \|^2 + \underbrace{4d\kappa L^2 h^2 + \frac{\kappa d^2 \Delta^2}{h^2}}_{\sigma^2} . \tag{7}$$

Now substituting into Theorem 2.6 instead of $\delta \to \kappa_\beta L h^{\beta-1}$ from equation 6, $\rho \to 4d\kappa$ from equation 7 and $\sigma^2 \to 4d\kappa L^2 h^2 + \frac{\kappa d^2 \Delta^2}{h^2}$ from equation 7, we obtain convergence for the novel gradient-free method (see Algorithm 1) with $\rho_B = \max\{1, \frac{4d\kappa}{B}\}$:

$$\mathbb{E}[f(x_N)] - f^* \leq \underbrace{\left( 1 - \sqrt{\frac{\mu}{\rho_B^2 L}} \right)^N \left[ f(x_0) - f^* + \frac{\mu}{2} \| x_0 - x^* \|^2 \right]}_{\text{①}} + \underbrace{\frac{4d\kappa L^2 h^2}{\sqrt{\rho_B^2 \mu L B^2}}}_{\text{②}}$$

$$+ \underbrace{\frac{\kappa d^2 \Delta^2}{h^2 \sqrt{\rho_B^2 \mu L B^2}}}_{\text{③}} + \underbrace{\frac{\kappa_\beta^2 L^2 h^{2(\beta-1)}}{\sqrt{4\mu L}}}_{\text{④}} .$$

We are now ready to present the main result of this paper.

**Theorem 3.1.** *Let the function $f$ satisfy Assumptions 1.1 and 1.2, and let the Kernel approximation with zero-order oracle (see Definition 1.3) satisfy Assumptions 1.4 and 2.3–2.4, then the novel Zero-Order Accelerated Batched Stochastic Gradient Descent (see Algorithm 1) converges to the desired accuracy $\varepsilon$ at the following parameters*

*Case $B = 1$: with smoothing parameter $h \lesssim \varepsilon^{1/2} \mu^{1/4}$, after $N = \mathcal{O}\left( \sqrt{\frac{d^2 L}{\mu}} \log \frac{1}{\varepsilon} \right)$ successive iterations, $T = N \cdot B = \mathcal{O}\left( \sqrt{\frac{d^2 L}{\mu}} \log \frac{1}{\varepsilon} \right)$ oracle calls and at $\Delta \lesssim \frac{\varepsilon \mu^{1/2}}{\sqrt{d}}$ maximum noise level.*

*Case $1 < B < 4d\kappa$: with parameter $h \lesssim \varepsilon^{1/2} \mu^{1/4}$, after $N = \mathcal{O}\left( \sqrt{\frac{d^2 L}{B^2 \mu}} \log \frac{1}{\varepsilon} \right)$ successive iterations, $T = N \cdot B = \mathcal{O}\left( \sqrt{\frac{d^2 L}{\mu}} \log \frac{1}{\varepsilon} \right)$ oracle calls and at $\Delta \lesssim \frac{\varepsilon \mu^{1/2}}{\sqrt{d}}$ maximum noise level.*

*Case $B = 4d\kappa$: with smoothing parameter $h \lesssim \varepsilon^{1/2} \mu^{1/4}$, after $N = \mathcal{O}\left( \sqrt{\frac{L}{\mu}} \log \frac{1}{\varepsilon} \right)$ successive iterations, $T = N \cdot B = \mathcal{O}\left( \sqrt{\frac{d^2 L}{\mu}} \log \frac{1}{\varepsilon} \right)$ oracle calls and at $\Delta \lesssim \frac{\varepsilon \mu^{1/2}}{\sqrt{d}}$ maximum noise level.*

*Case $B > 4d\kappa$: with parameter $h \lesssim \left( \varepsilon \sqrt{\mu} \right)^{\frac{1}{2(\beta-1)}}$, after $N = \mathcal{O}\left( \sqrt{\frac{L}{\mu}} \log \frac{1}{\varepsilon} \right)$ successive iterations, $T = N \cdot B = \max\{ \tilde{\mathcal{O}}\left( \sqrt{\frac{d^2 L}{\mu}} \right), \tilde{\mathcal{O}}\left( \frac{d^2 \Delta^2}{(\varepsilon\mu)^{\frac{\beta}{\beta-1}}} \right) \}$ oracle calls and at $\Delta \lesssim \frac{(\varepsilon\sqrt{\mu})^{\frac{\beta}{2(\beta-1)}}}{d} B^{1/2}$ maximum noise level.*

As can be seen from Theorem 3.1, Algorithm 1 indeed improves the iteration complexity compared to previous works (see Table 1), reaching the optimal estimate in a class of algorithms based on first-order algorithms at batch size $B = 4d\kappa$. However, if we consider the case $B \in [1, 4d\kappa]$, then when the batch size increases from 1, the algorithm improves the convergence rate (without changing the oracle complexity), but achieves the same error floor. This is not very good, because the asymptote does not depend on either the batch size or the smoothness order of the function. However, if we take the batch size larger than $B > 4d\kappa$, we will significantly improve the maximal noise level by worsening the oracle complexity. That is, in the overbatching condition, the error floor depends on both the batch size and the smoothness order, which can play a critical role in real life. For a detailed proof, see Appendix D.

**Remark 3.2** (Convex case.). *It is not difficult to show that the results of Theorem 3.1 generalize to the convex case (see Assumption 1.2 with $\mu = 0$), preserving the same dependence on $B$, namely in the case $B \in [1; 4d\kappa]$ and $h \lesssim \varepsilon^{3/4}$ we have the following convergence estimates for Algorithm 1: $N = \mathcal{O}\left(\sqrt{\frac{d^2 L R^2}{B^2 \varepsilon}}\right); T = \mathcal{O}\left(\sqrt{\frac{d^2 L R^2}{\varepsilon}}\right)$ and $\Delta \lesssim \frac{\varepsilon^{3/2}}{\sqrt{d}}$. We can also observe that the optimal estimate of iteration complexity in the convex setup is achieved when $B = 4d\kappa$. Moreover, the maximum noise level behaves in a similar way:$N = \mathcal{O}\left(\sqrt{\frac{L R^2}{\varepsilon}}\right); T =$*
$$\max\left[\mathcal{O}\left(\sqrt{\frac{d^2 L R^2}{\varepsilon}}\right), \mathcal{O}\left(\frac{d^2 \Delta^2}{\varepsilon^{2 + \frac{2}{\beta-1}}}\right)\right] \text{ and } \Delta \lesssim \frac{\varepsilon^{\frac{3\beta+1}{4(\beta-1)}}}{d} B^{1/2}.$$ *It can be seen that if we take $\mu \sim \varepsilon$, the oracle complexity is the same in the worst case, and the maximum noise level is inferior depending on the order of smoothness compared to the strongly convex set (which is surprising).*

**Remark 3.3** (Deterministic adversarial noise). *It should be noted that when considering deterministic adversarial noise ($|\tilde{\xi}(x)| \leq \Delta$) in a zero-order oracle instead of stochastic (see Definition 1.3), Theorem 3.1 will preserve the results except for the maximum noise level: $\Delta \lesssim \frac{(\varepsilon\sqrt{\mu})^{\frac{\beta}{2(\beta-1)}}}{d} B^{1/2} \rightarrow \Delta \lesssim \frac{(\varepsilon\sqrt{\mu})^{\frac{\beta}{2(\beta-1)}}}{d}$. This can be explained by the fact that deterministic noise is more adversarial because it accumulates not only in the second moment of the gradient approximation, but also in the bias! The results in the convex case will change similarly.*

**Remark 3.4** (High probability deviations bound). *Given that Algorithm 1 in strongly convex setting demonstrates a linear convergence rate and employs a randomization (see e.g. $\mathbf{e} \in S^d(1)$), we can derive exact estimates of high deviation probabilities using Markov's inequality Anikin et al. (2017):*

$$\mathcal{P}\left(f(x_{N_{(\varepsilon\omega)}}) - f^* \geq \varepsilon\right) \leq \omega \frac{\mathbb{E}\left[f(x_{N_{(\varepsilon\omega)}})\right] - f^*}{\varepsilon\omega} \leq \omega$$

.

**Remark 3.5** (Non-convex setup (PL)). *It should be noted that our algorithm will have global convergence for a subclass of non-convex functions that satisfy the Polyak—Lojasiewicz (PL) condition (see Karimi et al. (2016)). It is not hard to see that the results will have a similar dependence on the batch size: $N = \tilde{\mathcal{O}}\left(\frac{d}{B}\tilde{\mu}^{-1}\right); T = \tilde{\mathcal{O}}\left(d\tilde{\mu}^{-1}\right)$ and $\Delta \lesssim \frac{\varepsilon\tilde{\mu}}{\sqrt{d}}$, where $\tilde{\mu}$ from PL Assumption (see Karimi et al. (2016)). We can also observe that the optimal estimate of iteration complexity in the convex setup is achieved when $B = 4d\kappa$. Also, the maximum noise level behaves similarly: $N = \tilde{\mathcal{O}}\left(\tilde{\mu}^{-1}\right); T = \max\left[\tilde{\mathcal{O}}\left(d\tilde{\mu}^{-1}\right), \tilde{\mathcal{O}}\left(\frac{d^2 \Delta^2}{\varepsilon^{\frac{\beta}{\beta-1}}\tilde{\mu}^{\frac{2\beta-1}{\beta-1}}}\right)\right]$ and $\Delta \lesssim \frac{(\varepsilon\tilde{\mu})^{\frac{\beta}{2(\beta-1)}}}{d} B^{1/2}$.*

Similarly to the cases discussed above, when considering deterministic adversarial noise, the dependence on the batch size will disappear in the estimation of the maximum noise level. The transition to High probability deviations bounds is also valid. And if we compare with the estimates of Theorem 3.1, provided $\mu \sim \varepsilon$ from the strong convexity condition, and $\tilde{\mu} \sim \varepsilon$ from the PL condition, then the iteration complexity is the same, but the oracle complexity in the PL case is inferior to the strongly convex case. This can be explained by the fact that the PL condition covers a subclass of non-convex functions.

## 4 NUMERICAL EXPERIMENTS

In this section, we show the importance of considering the maximum noise level $\Delta$ as a third optimality criterion along with the standard two. We consider a problem of interest in machine learning, namely the logistic regression problem:

$$\min_{x \in \mathbb{R}^d}\left\{f(x) = \frac{1}{M}\sum_{i=1}^{M}\log(1 + \exp(-y_i \cdot (Ax)_i))\right\}.$$

Here we can understand $\log(1 + \exp(-y_i \cdot (Ax)_i)) = f_i(x)$ as the loss at the $i$-th data point, $x \in \mathbb{R}^d$ as a vector of parameters (or weights), $y \in \{-1, 1\}^M$ as a vector of labels, and $A \in \mathbb{R}^{M \times d}$ as a matrix of instances. For our experiments we use data from the LIBSVM library Chang & Lin (2011),

namely the a9a data. In the gradient approximation equation 5, we choose as the kernel function $K(r)$ the Legendre polynomials, for which it is shown in Bach & Perchet (2016) that the parameters $\kappa$ and $\kappa_\beta$ depend only on the smoothness order $\beta$. We have the following values for different $\beta$:

$$K(r) = \frac{15r}{4}(5 - 7r^2) \qquad\qquad \text{for } \beta = 3, 4;$$

$$K(r) = \frac{195r}{16}(99r^4 - 126r^2 + 35) \qquad\qquad \text{for } \beta = 5, 6.$$

To show the effectiveness of our Algorithm 1 (ZO-ABSGD) we compare with SOTA accelerated gradient-free algorithms, namely ZO-VARAG from Chen et al. (2020), ARDFDS from Gorbunov et al. (2022). We also compare our Algorithm 1 with RDFDS from Gorbunov et al. (2022) to demonstrate the superiority of the accelerated algorithm over the unaccelerated ones, which are all previous works (see Table 1).

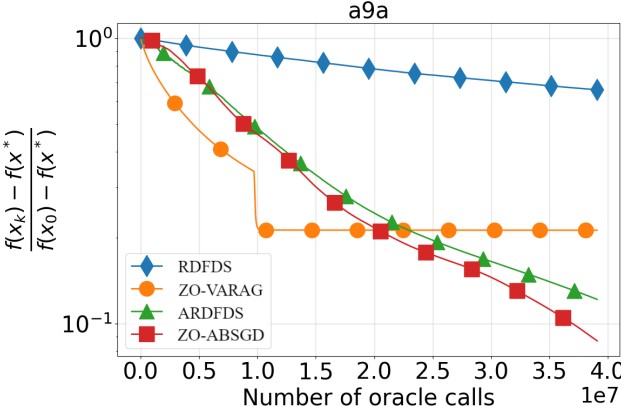

Figure 3: Comparison of SOTA gradient-free algorithms convergence. Here we optimize $f(x)$ with the parameters: $d = 123$ (problem dimensionality), $B = 1000$ (batch size), $\Delta = 10^{-5}$ (noise level), $\eta = 10^{-4}$ (step size), $h = 10^{-4}$ (smoothing parameter). In all experiments, the hyperparameters of the algorithms are tuned.

Figure 3 shows both standard results, such as the superiority of accelerated methods over unaccelerated methods, and the outperformance, the robustness of our algorithm. It is not hard to see that the ZO-VARAG algorithm outperforms the convergence rate on the first iterations, but converges to an error floor thereafter. This effect (convergence to the asymptote) can be explained by the fact that in Chen et al. (2020) an accelerated ZO-VARAG algorithm was proposed, which is not robust to adversarial noise. Regarding the RDFDS and ARDFDS algorithms, as the Figure shows they are also robust to adversarial stochastic noise like our algorithm. The robust convergence of the algorithms from Gorbunov et al. (2022) can be explained by the fact that in Gorbunov et al. (2022) algorithms were proposed that are robust to deterministic adversarial noise (DAN). As we know DAN is more antagonistic than stochastic adversarial noise because it accumulates not only in the variance but also in the bias of the gradient approximation. Despite this, ZO-ABSGD has better convergence compared to ARDFDS because the proposed 1 takes advantage of increased smoothness ($\beta = 3$), unlike its counterpart. Thus, this Figure 3 demonstrates not only the advantage of our algorithm, but also the importance in the design and analysis of algorithms robust to adversarial noise!

## 5 CONCLUSION

In this paper, we proposed a novel accelerated gradient-free algorithm to solve the black-box optimization problem under the assumption of increased smoothness and strong convexity of the objective function. By choosing a first-order accelerated algorithm and generalizing it to the Batched algorithm with a biased gradient oracle, we were able to improve the iteration complexity, reaching optimal estimates. Moreover, we have shown the importance of considering the maximum noise level as a third optimality criterion in a numerical experiment of interest in machine learning. And finally, we believe that this work offers a new perspective on black-box optimization and opens avenues for future research.

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

# APPENDIX

## A  AUXILIARY FACTS AND RESULTS

In this section we list auxiliary facts and results that we use several times in our proofs.

### A.1  SQUARED NORM OF THE SUM

For all $a_1, ..., a_n \in \mathbb{R}^d$, where $n = \{2, 3\}$

$$\|a_1 + ... + a_n\|^2 \leq n \|a_1\|^2 + ... + n \|a_n\|^2. \tag{8}$$

### A.2  FENCHEL-YOUNG INEQUALITY

For all $a, b \in \mathbb{R}^d$ and $\lambda > 0$

$$\langle a, b \rangle \leq \frac{\|a\|^2}{2\lambda} + \frac{\lambda \|b\|^2}{2}. \tag{9}$$

### A.3  $L$ SMOOTHNESS FUNCTION

Function $f$ is called $L$-smooth on $\mathbb{R}^d$ with $L > 0$ when it is differentiable and its gradient is $L$-Lipschitz continuous on $\mathbb{R}^d$, i.e.

$$\|\nabla f(x) - \nabla f(y)\| \leq L \|x - y\|, \quad \forall x, y \in \mathbb{R}^d. \tag{10}$$

It is well-known that $L$-smoothness implies (see e.g., Assumption 2.1)

$$f(y) \leq f(x) + \langle \nabla f(x), y - x \rangle + \frac{L}{2} \|y - x\|^2 \quad \forall x, y \in \mathbb{R}^d,$$

and if $f$ is additionally convex, then

$$\|\nabla f(x) - \nabla f(y)\|^2 \leq 2L \left( f(x) - f(y) - \langle \nabla f(y), x - y \rangle \right) \quad \forall x, y \in \mathbb{R}^d.$$

### A.4  WIRTINGER-POINCARE INEQUALITY

Let $f$ is differentiable, then for all $x \in \mathbb{R}^d$, $h\mathbf{e} \in S_2^d(h)$:

$$\mathbb{E}\left[ f(x + h\mathbf{e})^2 \right] \leq \frac{h^2}{d} \mathbb{E}\left[ \|\nabla f(x + h\mathbf{e})\|^2 \right]. \tag{11}$$

### A.5  TAYLOR EXPANSION

Using the Taylor expansion we have

$$\nabla_z f(x + hr\mathbf{u}) = \nabla_z f(x) + \sum_{1 \leq |n| \leq l-1} \frac{(rh)^{|n|}}{n!} D^{(n)} \nabla_z f(x) \mathbf{u}^n + R(hr\mathbf{u}), \tag{12}$$

where by assumption

$$|R(hr\mathbf{u})| \leq \frac{L}{(l-1)!} \|hr\mathbf{u}\|^{\beta-1} = \frac{L}{(l-1)!} |r|^{\beta-1} h^{\beta-1} \|\mathbf{u}\|^{\beta-1}. \tag{13}$$

### A.6 Kernel property

If $\mathbf{e}$ is uniformly distributed on $S_2^d(1)$ we have $\mathbb{E}[\mathbf{e}\mathbf{e}^{\mathsf{T}}] = (1/d)I_{d\times d}$, where $I_{d\times d}$ is the identity matrix. Therefore, using the facts $\mathbb{E}[rK(r)] = 1$ and $\mathbb{E}[r^{|n|}K(r)] = 0$ for $2 \le |n| \le l$ we have

$$\mathbb{E}\left[\frac{d}{h}\left(\langle \nabla f(x), hr\mathbf{e}\rangle + \sum_{2\le|n|\le l}\frac{(rh)^{|n|}}{n!}D^{(n)}f(x)\mathbf{e}^n\right)K(r)\mathbf{e}\right] = \nabla f(x). \tag{14}$$

### A.7 Bounds of the Weighted Sum of Legendre Polynomials

Let $\kappa_\beta = \int |u|^\beta |K(u)|du$ and set $\kappa = \int K^2(u)du$. Then if $K$ be a weighted sum of Legendre polynomials, then it is proved in (see Appendix A.3, Bach & Perchet (2016)) that $\kappa_\beta$ and $\kappa$ do not depend on $d$, they depend only on $\beta$, such that for $\beta \ge 1$:

$$\kappa_\beta \le 2\sqrt{2}(\beta - 1), \tag{15}$$

$$\kappa \le 3\beta^3. \tag{16}$$

## B Missing proof of Theorem 2.6

In this Section we demonstrate a missing proof of Theorem 2.6, namely a generalization of Lemma 2.5 to the case with a biased gradient oracle (see Definition 2.2). Therefore, our reasoning is based on the proof of Lemma 2.5 Vaswani et al. (2019).

Before proceeding directly to the proof, we recall the update rules of First-order Accelerated SGD from Vaswani et al. (2019):

$$y_k = \alpha_k z_k + (1 - \alpha_k)x_k; \tag{17}$$

$$x_{k+1} = y_k - \eta\mathbf{g}_k; \tag{18}$$

$$z_{k+1} = \beta_k z_k + (1 - \beta_k)y_k - \gamma_k\eta\mathbf{g}_k, \tag{19}$$

where we choose the parameters $\gamma_k, \alpha_k, \beta_k, a_k, b_k$ such that the following equations are satisfied:

$$\gamma_k = \frac{1}{2\rho}\cdot\left[1 + \frac{\beta_k(1 - \alpha_k)}{\alpha_k}\right]; \tag{20}$$

$$\alpha_k = \frac{\gamma_k\beta_k b_{k+1}^2\eta}{\gamma_k\beta_k b_{k+1}^2\eta + 2a_k^2}; \tag{21}$$

$$\beta_k \ge 1 - \gamma_k\mu\eta; \tag{22}$$

$$a_{k+1} = \gamma_k\sqrt{\eta\rho}b_{k+1}; \tag{23}$$

$$b_{k+1} \le \frac{b_k}{\sqrt{\beta_k}}. \tag{24}$$

Now, we're ready to move on to the proof itself. Let $r_{k+1} = \|z_{k+1} - x^*\|$ and $\mathbf{g}_k = \mathbf{g}(y_k, \xi_k)$ from Definition 2.2, then using equation equation 19:

$$r_{k+1}^2 = \|\beta_k\mathbf{k} + (1 - \beta_k)y_k - x^* - \gamma_k\eta\mathbf{g}_k\|^2$$

$$r_{k+1}^2 = \|\beta_k\mathbf{k} + (1 - \beta_k)y_k - x^*\|^2 + \gamma_k^2\eta^2\|\mathbf{g}_k\|^2 + 2\gamma_k\eta\langle x^* - \beta_k\mathbf{k} - (1 - \beta_k)y_k, \mathbf{g}_k\rangle.$$

Taking expectation wrt to $\xi_k$,

$$\mathbb{E}[r_{k+1}^2] = \mathbb{E}[\|\beta_k\mathbf{k} + (1 - \beta_k)y_k - x^*\|^2] + \gamma_k^2\eta^2\mathbb{E}\|\mathbf{g}_k\|^2$$
$$+ 2\gamma_k\eta\mathbb{E}\left[\langle x^* - \beta_k\mathbf{k} - (1 - \beta_k)y_k, \mathbf{g}_k\rangle\right]$$
$$\overset{equation\ 2.4}{\le} \|\beta_k\mathbf{k} + (1 - \beta_k)y_k - x^*\|^2 + \gamma_k^2\eta^2\rho\|\nabla f(y_k)\|^2$$
$$+ 2\gamma_k\eta\langle x^* - \beta_k\mathbf{k} - (1 - \beta_k)y_k, \mathbb{E}[\mathbf{g}_k]\rangle + \gamma_k^2\eta^2\sigma^2$$

$$
= \|\beta_k(\boldsymbol{k} - x^*) + (1 - \beta_k)(y_k - x^*)\|^2 + \gamma_k^2 \eta^2 \rho \|\nabla f(y_k)\|^2
$$
$$
+ 2\gamma_k \eta \langle x^* - \beta_k \boldsymbol{k} - (1 - \beta_k)y_k, \mathbb{E}[\mathbf{g}_k]\rangle + \gamma_k^2 \eta^2 \sigma^2
$$
$$
\leq \beta_k \|\boldsymbol{k} - x^*\|^2 + (1 - \beta_k)\|y_k - x^*\|^2 + \gamma_k^2 \eta^2 \rho \|\nabla f(y_k)\|^2
$$
$$
+ 2\gamma_k \eta \langle x^* - \beta_k \boldsymbol{k} - (1 - \beta_k)y_k, \mathbb{E}[\mathbf{g}_k]\rangle + \gamma_k^2 \eta^2 \sigma^2 \qquad \text{(By convexity of } \|\cdot\|^2\text{)}
$$
$$
= \beta_k \mathbf{k}^2 + (1 - \beta_k)\|y_k - x^*\|^2 + \gamma_k^2 \eta^2 \rho \|\nabla f(y_k)\|^2
$$
$$
+ 2\gamma_k \eta \langle x^* - \beta_k \boldsymbol{k} - (1 - \beta_k)y_k, \mathbb{E}[\mathbf{g}_k]\rangle + \gamma_k^2 \eta^2 \sigma^2
$$
$$
= \beta_k \mathbf{k}^2 + (1 - \beta_k)\|y_k - x^*\|^2 + \gamma_k^2 \eta^2 \rho \|\nabla f(y_k)\|^2
$$
$$
+ 2\gamma_k \eta \langle \beta_k(y_k - \boldsymbol{k}) + x^* - y_k, \mathbb{E}[\mathbf{g}_k]\rangle + \gamma_k^2 \eta^2 \sigma^2
$$
$$
\overset{\text{equation } 17}{=} \beta_k \mathbf{k}^2 + (1 - \beta_k)\|y_k - x^*\|^2 + \gamma_k^2 \eta^2 \rho \|\nabla f(y_k)\|^2
$$
$$
+ 2\gamma_k \eta \left\langle \frac{\beta_k(1 - \alpha_k)}{\alpha_k}(x_k - y_k) + x^* - y_k, \mathbb{E}[\mathbf{g}_k] \right\rangle + \gamma_k^2 \eta^2 \sigma^2
$$
$$
= \beta_k \mathbf{k}^2 + (1 - \beta_k)\|y_k - x^*\|^2 + \gamma_k^2 \eta^2 \rho \|\nabla f(y_k)\|^2
$$
$$
+ 2\gamma_k \eta \left[ \frac{\beta_k(1 - \alpha_k)}{\alpha_k}\langle \mathbb{E}[\mathbf{g}_k], (x_k - y_k)\rangle + \langle \mathbb{E}[\mathbf{g}_k], x^* - y_k\rangle \right]
$$
$$
+ \gamma_k^2 \eta^2 \sigma^2
$$
$$
\leq \beta_k \mathbf{k}^2 + (1 - \beta_k)\|y_k - x^*\|^2 + \gamma_k^2 \eta^2 \rho \|\nabla f(y_k)\|^2
$$
$$
+ 2\gamma_k \eta \left[ \frac{\beta_k(1 - \alpha_k)}{\alpha_k}(f(x_k) - f(y_k)) + \langle \mathbb{E}[\mathbf{g}_k], x^* - y_k\rangle \right] + \gamma_k^2 \eta^2 \sigma^2
$$
$$
+ 2\gamma_k \eta \left[ \frac{\beta_k(1 - \alpha_k)}{\alpha_k}\langle \mathbb{E}[\mathbf{g}_k] - \nabla f(y_k), x_k - y_k\rangle \right]. \qquad \text{(By convexity)}
$$

By strong convexity,

$$
\mathbb{E}[r_{k+1}^2] \leq \beta_k \mathbf{k}^2 + (1 - \beta_k)\|y_k - x^*\|^2 + \gamma_k^2 \eta^2 \rho \|\nabla f(y_k)\|^2
$$
$$
+ 2\gamma_k \eta \left[ \frac{\beta_k(1 - \alpha_k)}{\alpha_k}(f(x_k) - f(y_k)) + f^* - f(y_k) - \frac{\mu}{2}\|y_k - x^*\|^2 \right]
$$
$$
+ 2\gamma_k \eta \left[ \frac{\beta_k(1 - \alpha_k)}{\alpha_k}\langle \mathbb{E}[\mathbf{g}_k] - \nabla f(y_k), x_k - y_k\rangle + \langle \mathbb{E}[\mathbf{g}_k] - \nabla f(y_k), x^* - y_k\rangle \right]
$$
$$
+ \gamma_k^2 \eta^2 \sigma^2. \tag{25}
$$

By Lipschitz continuity of the gradient,

$$
f(x_{k+1}) - f(y_k) \leq \langle \nabla f(y_k), x_{k+1} - y_k\rangle + \frac{L}{2}\|x_{k+1} - y_k\|^2
$$
$$
\leq -\eta\langle \nabla f(y_k), \mathbf{g}_k\rangle + \frac{L\eta^2}{2}\|\mathbf{g}_k\|^2
$$
$$
= -\eta\|\nabla f(y_k)\|^2 + \frac{L\eta^2}{2}\|\mathbf{g}_k\|^2 - \eta\langle \nabla f(y_k), \mathbf{g}_k - \nabla f(y_k)\rangle.
$$

Taking expectation wrt $\xi_k$, we obtain

$$
\mathbb{E}[f(x_{k+1}) - f(y_k)] \leq -\eta\|\nabla f(y_k)\|^2 + \frac{L\rho\eta^2}{2}\|\nabla f(y_k)\|^2 + \frac{L\eta^2\sigma^2}{2}
$$
$$
- \eta\langle \nabla f(y_k), \mathbb{E}[\mathbf{g}_k] - \nabla f(y_k)\rangle
$$
$$
\mathbb{E}[f(x_{k+1}) - f(y_k)] \overset{\text{equation } 9}{\leq} \left[ -\frac{\eta}{2} + \frac{L\rho\eta^2}{2} \right]\|\nabla f(y_k)\|^2 + \frac{L\eta^2\sigma^2}{2}
$$
$$
+ \frac{\eta}{2}\|\mathbb{E}[\mathbf{g}_k] - \nabla f(y_k)\|^2.
$$

If $\eta \leq \frac{1}{2\rho L}$,

$$\mathbb{E}[f(x_{k+1}) - f(y_k)] \leq \left(\frac{-\eta}{4}\right) \|\nabla f(y_k)\|^2 + \frac{L\eta^2\sigma^2}{2} + \frac{\eta}{2} \|\mathbb{E}[\mathbf{g}_k] - \nabla f(y_k)\|^2$$

$$\|\nabla f(y_k)\|^2 \leq \left(\frac{4}{\eta}\right) \mathbb{E}[f(y_k) - f(x_{k+1})] + 2L\eta\sigma^2 + 2 \|\mathbb{E}[\mathbf{g}_k] - \nabla f(y_k)\|^2. \tag{26}$$

From equations equation 25 and equation 26, we get

$$\mathbb{E}[r_{k+1}^2] \leq \beta_k \mathbf{k}^2 + (1 - \beta_k) \|y_k - x^*\|^2 + 4\gamma_k^2 \rho \eta \mathbb{E}[f(y_k) - f(x_{k+1})]$$
$$+ 2\gamma_k \eta \left[ \frac{\beta_k(1 - \alpha_k)}{\alpha_k} (f(x_k) - f(y_k)) + f^* - f(y_k) - \frac{\mu}{2} \|y_k - x^*\|^2 \right]$$
$$+ \left[ 2\gamma_k \eta \cdot \frac{\beta_k(1 - \alpha_k)}{\alpha_k} \right] \langle \mathbb{E}[\mathbf{g}_k] - \nabla f(y_k), x_k - y_k \rangle$$
$$+ 2\gamma_k \eta \langle \mathbb{E}[\mathbf{g}_k] - \nabla f(y_k), x^* - y_k \rangle$$
$$+ \gamma_k^2 \eta^2 \sigma^2 + 2L\gamma_k^2 \eta^3 \rho \sigma^2 + 2\gamma_k^2 \eta^2 \rho \|\mathbb{E}[\mathbf{g}_k] - \nabla f(y_k)\|^2$$
$$\leq \beta_k \mathbf{k}^2 + (1 - \beta_k) \|y_k - x^*\|^2 + 4\gamma_k^2 \eta \rho \mathbb{E}[f(y_k) - f(x_{k+1})]$$
$$+ 2\gamma_k \eta \left[ \frac{\beta_k(1 - \alpha_k)}{\alpha_k} (f(x_k) - f(y_k)) + f^* - f(y_k) - \frac{\mu}{2} \|y_k - x^*\|^2 \right]$$
$$+ \left[ 2\gamma_k \eta \cdot \frac{\beta_k(1 - \alpha_k)}{\alpha_k} \right] \langle \mathbb{E}[\mathbf{g}_k] - \nabla f(y_k), x_k - y_k \rangle$$
$$+ 2\gamma_k \eta \langle \mathbb{E}[\mathbf{g}_k] - \nabla f(y_k), x^* - y_k \rangle$$
$$+ 2\gamma_k^2 \eta^2 \sigma^2 + 2\gamma_k^2 \eta^2 \rho \|\mathbb{E}[\mathbf{g}_k] - \nabla f(y_k)\|^2 \qquad \text{(Since } \eta \leq \frac{1}{\rho L})$$
$$= \beta_k \mathbf{k}^2 + \|y_k - x^*\|^2 [(1 - \beta_k) - \gamma_k \mu \eta]$$
$$+ f(y_k) \left[ 4\gamma_k^2 \eta \rho - 2\gamma_k \eta \cdot \frac{\beta_k(1 - \alpha_k)}{\alpha_k} - 2\gamma_k \eta \right]$$
$$- 4\gamma_k^2 \eta \rho \mathbb{E} f(x_{k+1}) + 2\gamma_k \eta f^* + \left[ 2\gamma_k \eta \cdot \frac{\beta_k(1 - \alpha_k)}{\alpha_k} \right] f(x_k)$$
$$+ \left[ 2\gamma_k \eta \cdot \frac{\beta_k(1 - \alpha_k)}{\alpha_k} \right] \langle \mathbb{E}[\mathbf{g}_k] - \nabla f(y_k), x_k - y_k \rangle$$
$$+ 2\gamma_k \eta \langle \mathbb{E}[\mathbf{g}_k] - \nabla f(y_k), x^* - y_k \rangle$$
$$+ 2\gamma_k^2 \eta^2 \sigma^2 + 2\gamma_k^2 \eta^2 \rho \|\mathbb{E}[\mathbf{g}_k] - \nabla f(y_k)\|^2.$$

Since $\beta_k \geq 1 - \gamma_k \mu \eta$ and $\gamma_k = \frac{1}{2\rho} \cdot \left(1 + \frac{\beta_k(1 - \alpha_k)}{\alpha_k}\right)$,

$$\mathbb{E}[r_{k+1}^2] \leq \beta_k \mathbf{k}^2 - 4\gamma_k^2 \eta \rho \mathbb{E} f(x_{k+1}) + 2\gamma_k \eta f^* + \left[ 2\gamma_k \eta \cdot \frac{\beta_k(1 - \alpha_k)}{\alpha_k} \right] f(x_k)$$
$$+ \left[ 2\gamma_k \eta \cdot \frac{\beta_k(1 - \alpha_k)}{\alpha_k} \right] \langle \mathbb{E}[\mathbf{g}_k] - \nabla f(y_k), x_k - y_k \rangle$$
$$+ 2\gamma_k \eta \langle \mathbb{E}[\mathbf{g}_k] - \nabla f(y_k), x^* - y_k \rangle$$
$$+ 2\gamma_k^2 \eta^2 \sigma^2 + 2\gamma_k^2 \eta^2 \rho \|\mathbb{E}[\mathbf{g}_k] - \nabla f(y_k)\|^2.$$

Multiplying by $b_{k+1}^2$,

$$b_{k+1}^2 \mathbb{E}[r_{k+1}^2] \leq b_{k+1}^2 \beta_k \mathbf{k}^2 - 4b_{k+1}^2 \gamma_k^2 \eta \rho \mathbb{E} f(x_{k+1}) + 2b_{k+1}^2 \gamma_k \eta f^*$$
$$+ \left[ 2b_{k+1}^2 \gamma_k \eta \cdot \frac{\beta_k(1 - \alpha_k)}{\alpha_k} \right] f(x_k)$$
$$+ \left[ 2b_{k+1}^2 \gamma_k \eta \cdot \frac{\beta_k(1 - \alpha_k)}{\alpha_k} \right] \langle \mathbb{E}[\mathbf{g}_k] - \nabla f(y_k), x_k - y_k \rangle$$

$$+ 2b_{k+1}^2 \gamma_k \eta \left\langle \mathbb{E}\left[\mathbf{g}_k\right] - \nabla f(y_k), x^* - y_k \right\rangle$$
$$+ 2b_{k+1}^2 \gamma_k^2 \eta^2 \sigma^2 + 2b_{k+1}^2 \gamma_k^2 \eta^2 \rho \left\| \mathbb{E}\left[\mathbf{g}_k\right] - \nabla f(y_k) \right\|^2.$$

Since $b_{k+1}^2 \beta_k \leq b_k^2$, $b_{k+1}^2 \gamma_k^2 \eta \rho = a_{k+1}^2$, $\frac{\gamma_k \eta \beta_k (1-\alpha_k)}{\alpha_k} = \frac{2a_k^2}{b_{k+1}^2}$

$$b_{k+1}^2 \mathbb{E}[r_{k+1}^2] \leq b_k^2 \mathsf{k}^2 - 4a_{k+1}^2 \mathbb{E}f(x_{k+1}) + 2b_{k+1}^2 \gamma_k \eta f^* + 4a_k^2 f(x_k)$$
$$+ 4a_k^2 \left\langle \mathbb{E}\left[\mathbf{g}_k\right] - \nabla f(y_k), x_k - y_k \right\rangle$$
$$+ 2b_{k+1}^2 \gamma_k \eta \left\langle \mathbb{E}\left[\mathbf{g}_k\right] - \nabla f(y_k), x^* - y_k \right\rangle$$
$$+ \frac{2a_{k+1}^2 \sigma^2 \eta}{\rho} + 2a_{k+1}^2 \eta \left\| \mathbb{E}\left[\mathbf{g}_k\right] - \nabla f(y_k) \right\|^2$$
$$= b_k^2 \mathsf{k}^2 - 4a_{k+1}^2 \left[\mathbb{E}f(x_{k+1}) - f^*\right] + 4a_k^2 \left[f(x_k) - f^*\right]$$
$$+ 2 \left[b_{k+1}^2 \gamma_k \eta - 2a_{k+1}^2 + 2a_k^2\right] f^*$$
$$+ 4a_k^2 \left\langle \mathbb{E}\left[\mathbf{g}_k\right] - \nabla f(y_k), x_k - y_k \right\rangle$$
$$+ 2b_{k+1}^2 \gamma_k \eta \left\langle \mathbb{E}\left[\mathbf{g}_k\right] - \nabla f(y_k), x^* - y_k \right\rangle$$
$$+ \frac{2a_{k+1}^2 \sigma^2 \eta}{\rho} + 2a_{k+1}^2 \eta \left\| \mathbb{E}\left[\mathbf{g}_k\right] - \nabla f(y_k) \right\|^2.$$

Since $\left[b_{k+1}^2 \gamma_k \eta - a_{k+1}^2 + a_k^2\right] = 0$,

$$b_{k+1}^2 \mathbb{E}[r_{k+1}^2] \leq b_k^2 \mathsf{k}^2 - 4a_{k+1}^2 \left[\mathbb{E}f(x_{k+1}) - f^*\right] + 4a_k^2 \left[f(x_k) - f^*\right]$$
$$+ 4a_k^2 \left\langle \mathbb{E}\left[\mathbf{g}_k\right] - \nabla f(y_k), x_k - x^* \right\rangle$$
$$+ 4a_{k+1}^2 \left\langle \mathbb{E}\left[\mathbf{g}_k\right] - \nabla f(y_k), x^* - y_k \right\rangle$$
$$+ \frac{2a_{k+1}^2 \sigma^2 \eta}{\rho} + 2a_{k+1}^2 \eta \left\| \mathbb{E}\left[\mathbf{g}_k\right] - \nabla f(y_k) \right\|^2.$$

Denoting $\mathbb{E}f(x_{k+1}) - f^*$ as $\Phi_{k+1}$, we obtain

$$4a_{k+1}^2 \Phi_{k+1} - 4a_k^2 \Phi_k \overset{equation\ 2.3}{\leq} b_k^2 \mathsf{k}^2 - b_{k+1}^2 \mathbb{E}[r_{k+1}^2]$$
$$+ 4a_k^2 \delta \tilde{R} - 4a_{k+1}^2 \delta \tilde{R}$$
$$+ \frac{2a_{k+1}^2 \sigma^2 \eta}{\rho} + 2a_{k+1}^2 \eta \delta^2,$$

where $\tilde{R} = \max_k \{\|x_k - x^*\|, \|y_k - x^*\|\}$.

By summing over $k$ we obtain:

$$4 \sum_{k=0}^{N-1} \left[a_{k+1}^2 \Phi_{k+1} - a_k^2 \Phi_k\right] \leq \sum_{k=0}^{N-1} \left[b_k^2 \mathsf{k}^2 - b_{k+1}^2 \mathbb{E}[r_{k+1}^2]\right]$$
$$+ 4 \sum_{k=0}^{N-1} \left[a_k^2 \delta \tilde{R} - a_{k+1}^2 \delta \tilde{R}\right]$$
$$+ \sum_{k=0}^{N-1} \left[\frac{2a_{k+1}^2 \sigma^2 \eta}{\rho}\right] + 2 \sum_{k=0}^{N-1} \left[a_{k+1}^2 \eta \delta^2\right].$$

Let's substitute $a_{k+1}^2 = b_{k+1}^2 \gamma_k^2 \eta \rho$:

$$4b_N^2 \gamma_{N-1}^2 \eta \rho \Phi_N \leq 4a_0^2 \Phi_0 + b_0^2 r_0^2 - b_N^2 \mathbb{E}\left[r_N^2\right]$$
$$+ 4a_0^2 \delta \tilde{R} - 4a_N^2 \delta \tilde{R}$$

$$+ \sum_{k=0}^{N-1} \left[ \frac{2a_{k+1}^2 \sigma^2 \eta}{\rho} \right] + 2 \sum_{k=0}^{N-1} \left[ a_{k+1}^2 \eta \delta^2 \right].$$

Divide the left and right parts by $4\rho\eta$:

$$b_N^2 \gamma_{N-1}^2 \Phi_N \leq \frac{a_0^2}{\rho\eta} \Phi_0 + \frac{b_0^2 r_0^2}{4\rho\eta} + \frac{a_0^2 \tilde{R}}{\rho\eta} \delta + \frac{\eta\sigma^2}{2\rho} \sum_{k=0}^{N-1} \left[ b_{k+1}^2 \gamma_k^2 \right] + \frac{\eta}{2} \delta^2 \sum_{k=0}^{N-1} \left[ b_{k+1}^2 \gamma_k^2 \right].$$

Next, we show that according to equation 20-equation 24 the following relation is correct:

$$\gamma_k^2 - \gamma_k \left[ \frac{1}{2\rho} - \mu\eta\gamma_{k-1}^2 \right] = \gamma_{k-1}^2$$

Namely,

$$\gamma_k \overset{equation\ 20}{=} \frac{1}{2\rho} \left[ 1 + \frac{\beta_k(1 - \alpha_k)}{\alpha_k} \right]$$

$$\gamma_k^2 - \frac{\gamma_k}{2\rho} = \frac{\gamma_k \beta_k (1 - \alpha_k)}{2\rho\alpha_k}$$

$$\overset{equation\ 21}{=} \frac{1}{\eta\rho} \frac{a_k^2}{b_{k+1}^2}$$

$$\overset{equation\ 24}{=} \frac{\beta_k}{\eta\rho} \frac{a_k^2}{b_k^2}$$

$$\overset{equation\ 22}{=} \frac{1 - \gamma_k \mu\eta}{\eta\rho} \frac{a_k^2}{b_k^2}$$

$$\overset{equation\ 23}{=} \frac{1 - \gamma_k \mu\eta}{\eta\rho} \left( \gamma_{k-1} \sqrt{\eta\rho} \right)^2$$

$$= (1 - \gamma_k \mu\eta) \gamma_{k-1}^2$$

$$\Rightarrow \quad \gamma_k^2 - \gamma_k \left[ \frac{1}{2\rho} - \mu\eta\gamma_{k-1}^2 \right] = \gamma_{k-1}^2. \tag{27}$$

If $\gamma_k = C$, then

$$\gamma_k = \frac{1}{\sqrt{2\mu\eta\rho}}$$

$$\beta_k = 1 - \sqrt{\frac{\mu\eta}{2\rho}}$$

$$b_{k+1} = \frac{b_0}{\left( 1 - \sqrt{\frac{\mu\eta}{2\rho}} \right)^{(k+1)/2}}$$

$$a_{k+1} = \frac{1}{\sqrt{2\mu\eta\rho}} \cdot \sqrt{\eta\rho} \cdot \frac{b_0}{\left( 1 - \sqrt{\frac{\mu\eta}{2\rho}} \right)^{(k+1)/2}} = \frac{b_0}{\sqrt{2\mu}} \cdot \frac{1}{\left( 1 - \sqrt{\frac{\mu\eta}{2\rho}} \right)^{(k+1)/2}}.$$

If $b_0 = \sqrt{2\mu}$,

$$a_{k+1} = \frac{1}{\left( 1 - \sqrt{\frac{\mu\eta}{2\rho}} \right)^{(k+1)/2}}.$$

The above equation implies that $a_0 = 1$.

Now the above relations allow us to obtain the following inequality:

$$\frac{2\mu}{\left( 1 - \sqrt{\frac{\mu\eta}{2\rho}} \right)^N} \frac{1}{2\mu\eta\rho} \Phi_N \leq \frac{1}{\rho\eta} \Phi_0 + \frac{2\mu r_0^2}{4\rho\eta} + \frac{\tilde{R}}{\rho\eta} \delta$$

$$+ \frac{\sigma^2}{\rho^2} \sum_{k=0}^{N-1} \left[ \frac{1}{\left(1 - \sqrt{\frac{\mu\eta}{2\rho}}\right)^{(k+1)}} \right]$$

$$+ \frac{1}{2\rho} \delta^2 \sum_{k=0}^{N-1} \left[ \frac{1}{\left(1 - \sqrt{\frac{\mu\eta}{2\rho}}\right)^{(k+1)}} \right];$$

$$\frac{1}{\left(1 - \sqrt{\frac{\mu\eta}{2\rho}}\right)^N} \Phi_N \le \Phi_0 + \frac{\mu}{2} r_0^2 + \tilde{R}\delta$$

$$+ \frac{\sigma^2 \eta}{\rho} \sum_{k=0}^{N-1} \left[ \frac{1}{\left(1 - \sqrt{\frac{\mu\eta}{2\rho}}\right)^{(k+1)}} \right]$$

$$+ \frac{\eta}{2} \delta^2 \sum_{k=0}^{N-1} \left[ \frac{1}{\left(1 - \sqrt{\frac{\mu\eta}{2\rho}}\right)^{(k+1)}} \right];$$

$$\frac{1}{\left(1 - \sqrt{\frac{\mu\eta}{2\rho}}\right)^N} \Phi_N \le \Phi_0 + \frac{\mu}{2} r_0^2 + \tilde{R}\delta$$

$$+ \frac{\sigma^2 \sqrt{2\eta}}{\sqrt{\rho\mu}} \cdot \frac{1}{\left(1 - \sqrt{\frac{\mu\eta}{2\rho}}\right)^N}$$

$$+ \frac{\sqrt{\eta\rho}}{\sqrt{2\mu}} \delta^2 \cdot \frac{1}{\left(1 - \sqrt{\frac{\mu\eta}{2\rho}}\right)^{(k+1)}};$$

$$\mathbb{E}\left[f(x_N)\right] - f^* \le \left(1 - \sqrt{\frac{\mu\eta}{2\rho}}\right)^N \left[f(x_0) - f^* + \frac{\mu}{2} r_0^2\right]$$

$$+ \left(1 - \sqrt{\frac{\mu\eta}{2\rho}}\right)^N \tilde{R}\delta + \frac{\sigma^2 \sqrt{2\eta}}{\sqrt{\rho\mu}} + \frac{\sqrt{\eta\rho}}{\sqrt{2\mu}} \delta^2;$$

$$\mathbb{E}\left[f(x_N)\right] - f^* \le \left(1 - \sqrt{\frac{\mu}{4\rho^2 L}}\right)^N \left[f(x_0) - f^* + \frac{\mu}{2} \|x_0 - x^*\|^2\right]$$

$$+ \left(1 - \sqrt{\frac{\mu}{4\rho^2 L}}\right)^N \tilde{R}\delta + \frac{\sigma^2}{\sqrt{\rho^2 \mu L}} + \frac{1}{\sqrt{4\mu L}} \delta^2.$$

By adding batching, given that $\tilde{\rho}_B = \max\{1, \frac{\rho}{B}\}$ and $\sigma_B^2 = \frac{\sigma^2}{B}$ we have the convergence rate for accelerated batched SGD with biased gradient oracle and parameter $\eta \lesssim \frac{1}{2\tilde{\rho}_B L}$:

$$\mathbb{E}\left[f(x_N)\right] - f^* \le \left(1 - \sqrt{\frac{\mu}{4\tilde{\rho}_B^2 L}}\right)^N \left[f(x_0) - f^* + \frac{\mu}{2} \|x_0 - x^*\|^2\right]$$

$$+ \left(1 - \sqrt{\frac{\mu}{4\tilde{\rho}_B^2 L}}\right)^N \tilde{R}\delta + \frac{\sigma_B^2}{\sqrt{\tilde{\rho}_B^2 \mu L}} + \frac{1}{\sqrt{4\mu L}} \delta^2.$$

## C  PROPERTIES OF THE KERNEL APPROXIMATION

In this Section, we extend the explanations for obtaining the bias and second moment estimates of the gradient approximation.

Using the variational representation of the Euclidean norm, and definition of gradient approximation equation 5 we can write:

$$\|\mathbf{b}(x_k)\| = \|\mathbb{E}\left[\mathbf{g}(x_k, \mathbf{e})\right] - \nabla f(x_k)\|$$

$$= \left\|\frac{d}{2h}\mathbb{E}\left[\left(\tilde{f}(x_k + hr\mathbf{e}) - \tilde{f}(x_k - hr\mathbf{e})\right)K(r)\mathbf{e}\right] - \nabla f(x_k)\right\|$$

$$\overset{\text{①}}{=} \left\|\frac{d}{h}\mathbb{E}\left[f(x_k + hr\mathbf{e})K(r)\mathbf{e}\right] - \nabla f(x_k)\right\|$$

$$\overset{\text{②}}{=} \|\mathbb{E}\left[\nabla f(x_k + hr\mathbf{u})rK(r)\right] - \nabla f(x_k)\|$$

$$= \sup_{z \in S_2^d(1)} \mathbb{E}\left[(\nabla_z f(x_k + hr\mathbf{u}) - \nabla_z f(x_k))rK(r)\right]$$

$$\overset{\text{equation 12,equation 13}}{\leq} \kappa_\beta h^{\beta-1}\frac{L}{(l-1)!}\mathbb{E}\left[\|u\|^{\beta-1}\right]$$

$$\leq \kappa_\beta h^{\beta-1}\frac{L}{(l-1)!}\frac{d}{d+\beta-1}$$

$$\lesssim \kappa_\beta L h^{\beta-1},$$

where $u \in B^d(1)$, ① = the equality is obtained from the fact, namely, distribution of $\mathbf{e}$ is symmetric, ② = the equality is obtained from a version of Stokes' theorem Zorich & Paniagua (2016).

By definition gradient approximation equation 5 and Wirtinger-Poincare inequality equation 11 we have

$$\mathbb{E}\left[\|\mathbf{g}(x_k, \mathbf{e})\|^2\right] = \frac{d^2}{4h^2}\mathbb{E}\left[\left\|\left(\tilde{f}(x_k + hr\mathbf{e}) - \tilde{f}(x_k - hr\mathbf{e})\right)K(r)\mathbf{e}\right\|^2\right]$$

$$= \frac{d^2}{4h^2}\mathbb{E}\left[\left(f(x_k + hr\mathbf{e}) - f(x_k - hr\mathbf{e}) + (\xi_1 - \xi_2))\right)^2 K^2(r)\right]$$

$$\overset{\text{equation 8}}{\leq} \frac{\kappa d^2}{2h^2}\left(\mathbb{E}\left[(f(x_k + hr\mathbf{e}) - f(x_k - hr\mathbf{e}))^2\right] + 2\Delta^2\right)$$

$$\overset{\text{equation 11}}{\leq} \frac{\kappa d^2}{2h^2}\left(\frac{h^2}{d}\mathbb{E}\left[\|\nabla f(x_k + hr\mathbf{e}) + \nabla f(x_k - hr\mathbf{e})\|^2\right] + 2\Delta^2\right)$$

$$= \frac{\kappa d^2}{2h^2}\left(\frac{h^2}{d}\mathbb{E}\left[\|\nabla f(x_k + hr\mathbf{e}) + \nabla f(x_k - hr\mathbf{e}) \pm 2\nabla f(x_k)\|^2\right] + 2\Delta^2\right)$$

$$\overset{\text{equation 10}}{\leq} \underbrace{4d\kappa}_{\rho}\|\nabla f(x_k)\|^2 + \underbrace{4d\kappa L^2 h^2 + \frac{\kappa d^2\Delta^2}{h^2}}_{\sigma^2}.$$

## D   MISSING PROOF OF THEOREM 3.1

Let us consider case $B = 1$, then we have the following convergence rate:

$$\mathbb{E}\left[f(x_N)\right] - f^* \leq \underbrace{\left(1 - \sqrt{\frac{\mu}{(4d\kappa)^2 L}}\right)^N \left[f(x_0) - f^* + \frac{\mu}{2}\|x_0 - x^*\|^2\right]}_{\text{①}} + \underbrace{\frac{4d\kappa L^2 h^2}{\sqrt{(4d\kappa)^2\mu L}}}_{\text{②}}$$

$$+ \underbrace{\frac{\kappa d^2\Delta^2}{h^2\sqrt{(4d\kappa)^2\mu L}}}_{\text{③}} + \underbrace{\frac{\kappa_\beta^2 L^2 h^{2(\beta-1)}}{\sqrt{4\mu L}}}_{\text{④}}.$$

**From term** ①, we find iteration number $N$ required to achieve $\varepsilon$-accuracy:

$$\left(1 - \sqrt{\frac{\mu}{(4d\kappa)^2 L}}\right)^N \left[f(x_0) - f^* + \frac{\mu}{2}\|x_0 - x^*\|^2\right] \leq \varepsilon \quad \Rightarrow \quad \boxed{N = \tilde{\mathcal{O}}\left(\sqrt{\frac{d^2 L}{\mu}}\right).}$$

**From terms ②, ④** we find the smoothing parameter $h$:

$$② : \quad \frac{4d\kappa L^2 h^2}{\sqrt{(4d\kappa)^2 \mu L}} \leq \varepsilon \quad \Rightarrow \quad h^2 \lesssim \varepsilon\sqrt{\mu} \quad \Rightarrow \quad \boxed{h \lesssim (\varepsilon\sqrt{\mu})^{1/2};}$$

$$④ : \quad \frac{\kappa_\beta^2 L^2 h^{2(\beta-1)}}{\sqrt{4\mu L}} \leq \varepsilon \quad \Rightarrow \quad h^{2(\beta-1)} \lesssim \varepsilon\sqrt{\mu} \quad \Rightarrow \quad h \lesssim (\varepsilon\sqrt{\mu})^{\frac{1}{2(\beta-1)}}.$$

**From term ③**, we find the maximum noise level $\Delta$ at which Algorithm 1 can still achieve the desired accuracy:

$$\frac{\kappa d^2 \Delta^2}{h^2 \sqrt{(4d\kappa)^2 \mu L}} \leq \varepsilon \quad \Rightarrow \quad \Delta^2 \lesssim \frac{\varepsilon\sqrt{\mu}h^2}{d} \quad \Rightarrow \quad \boxed{\Delta \lesssim \frac{\varepsilon\sqrt{\mu}}{\sqrt{d}}.}$$

The oracle complexity in this case is obtained as follows:

$$\boxed{T = N \cdot B = \tilde{\mathcal{O}}\left(\sqrt{\frac{d^2 L}{\mu}}\right).}$$

Consider now the case $1 < B < 4d\kappa$, then we have the convergence rate:

$$\mathbb{E}\left[f(x_N)\right] - f^* \leq \underbrace{\left(1 - \sqrt{\frac{\mu B^2}{(4d\kappa)^2 L}}\right)^N \left[f(x_0) - f^* + \frac{\mu}{2}\|x_0 - x^*\|^2\right]}_{①} + \underbrace{\frac{4d\kappa L^2 h^2}{\sqrt{(4d\kappa)^2 \mu L}}}_{②}$$

$$+ \underbrace{\frac{\kappa d^2 \Delta^2}{h^2 \sqrt{(4d\kappa)^2 \mu L}}}_{③} + \underbrace{\frac{\kappa_\beta^2 L^2 h^{2(\beta-1)}}{\sqrt{4\mu L}}}_{④}.$$

**From term ①**, we find iteration number $N$ required for Algorithm 1 to achieve $\varepsilon$-accuracy:

$$\left(1 - \sqrt{\frac{B^2 \mu}{(4d\kappa)^2 L}}\right)^N \left[f(x_0) - f^* + \frac{\mu}{2}\|x_0 - x^*\|^2\right] \leq \varepsilon \quad \Rightarrow \quad \boxed{N = \tilde{\mathcal{O}}\left(\sqrt{\frac{d^2 L}{B^2 \mu}}\right).}$$

**From terms ②, ④** we find the smoothing parameter $h$:

$$② : \quad \frac{4d\kappa L^2 h^2}{\sqrt{(4d\kappa)^2 \mu L}} \leq \varepsilon \quad \Rightarrow \quad h^2 \lesssim \varepsilon\sqrt{\mu} \quad \Rightarrow \quad \boxed{h \lesssim (\varepsilon\sqrt{\mu})^{1/2};}$$

$$④ : \quad \frac{\kappa_\beta^2 L^2 h^{2(\beta-1)}}{\sqrt{4\mu L}} \leq \varepsilon \quad \Rightarrow \quad h^{2(\beta-1)} \lesssim \varepsilon\sqrt{\mu} \quad \Rightarrow \quad h \lesssim (\varepsilon\sqrt{\mu})^{\frac{1}{2(\beta-1)}}.$$

**From term ③**, we find the maximum noise level $\Delta$ at which Algorithm 1 can still achieve the desired accuracy:

$$\frac{\kappa d^2 \Delta^2}{h^2 \sqrt{(4d\kappa)^2 \mu L}} \leq \varepsilon \quad \Rightarrow \quad \Delta^2 \lesssim \frac{\varepsilon\sqrt{\mu}h^2}{d} \quad \Rightarrow \quad \boxed{\Delta \lesssim \frac{\varepsilon\sqrt{\mu}}{\sqrt{d}}.}$$

The oracle complexity in this case is obtained as follows:

$$\boxed{T = N \cdot B = \tilde{\mathcal{O}}\left(\sqrt{\frac{d^2 L}{\mu}}\right).}$$

Now let us move to the case where $B = 4d\kappa$, then we have convergence rate:

$$\mathbb{E}\left[f(x_N)\right] - f^* \leq \underbrace{\left(1 - \sqrt{\frac{\mu}{L}}\right)^N \left[f(x_0) - f^* + \frac{\mu}{2}\|x_0 - x^*\|^2\right]}_{①} + \underbrace{\frac{L^2 h^2}{\sqrt{\mu L}}}_{②}$$

$$+ \underbrace{\frac{d\Delta^2}{h^2\sqrt{\mu L}}}_{\text{③}} + \underbrace{\frac{\kappa_\beta^2 L^2 h^{2(\beta-1)}}{\sqrt{4\mu L}}}_{\text{④}}.$$

**From term ①,** we find iteration number $N$ required for Algorithm 1 to achieve $\varepsilon$-accuracy:

$$\left(1 - \sqrt{\frac{\mu}{L}}\right)^N \left[f(x_0) - f^* + \frac{\mu}{2}\|x_0 - x^*\|^2\right] \le \varepsilon \quad \Rightarrow \quad \boxed{N = \tilde{\mathcal{O}}\left(\sqrt{\frac{L}{\mu}}\right).}$$

**From terms ②, ④** we find the smoothing parameter $h$:

$$\text{②}: \quad \frac{L^2 h^2}{\sqrt{\mu L}} \le \varepsilon \quad \Rightarrow \quad h^2 \lesssim \varepsilon\sqrt{\mu} \quad \Rightarrow \quad \boxed{h \lesssim (\varepsilon\sqrt{\mu})^{1/2};}$$

$$\text{④}: \quad \frac{\kappa_\beta^2 L^2 h^{2(\beta-1)}}{\sqrt{4\mu L}} \le \varepsilon \quad \Rightarrow \quad h^{2(\beta-1)} \lesssim \varepsilon\sqrt{\mu} \quad \Rightarrow \quad h \lesssim (\varepsilon\sqrt{\mu})^{\frac{1}{2(\beta-1)}}.$$

**From term ③,** we find the maximum noise level $\Delta$ at which Algorithm 1 can still achieve the desired accuracy:

$$\frac{d\Delta^2}{h^2\sqrt{\mu L}} \le \varepsilon \quad \Rightarrow \quad \Delta^2 \lesssim \frac{\varepsilon\sqrt{\mu}h^2}{d} \quad \Rightarrow \quad \boxed{\Delta \lesssim \frac{\varepsilon\sqrt{\mu}}{\sqrt{d}}.}$$

The oracle complexity in this case is obtained as follows:

$$\boxed{T = N \cdot B = \tilde{\mathcal{O}}\left(\sqrt{\frac{d^2 L}{\mu}}\right).}$$

Finally, consider the case when $B > 4d\kappa$, then we have convergence rate:

$$\mathbb{E}[f(x_N)] - f^* \le \underbrace{\left(1 - \sqrt{\frac{\mu}{L}}\right)^N \left[f(x_0) - f^* + \frac{\mu}{2}\|x_0 - x^*\|^2\right]}_{\text{①}} + \underbrace{\frac{4d\kappa L^2 h^2}{\sqrt{\mu L B^2}}}_{\text{②}}$$

$$+ \underbrace{\frac{\kappa d^2 \Delta^2}{h^2\sqrt{\mu L B^2}}}_{\text{③}} + \underbrace{\frac{\kappa_\beta^2 L^2 h^{2(\beta-1)}}{\sqrt{4\mu L}}}_{\text{④}}.$$

**From term ①,** we find iteration number $N$ required for Algorithm 1 to achieve $\varepsilon$-accuracy:

$$\left(1 - \sqrt{\frac{\mu}{L}}\right)^N \left[f(x_0) - f^* + \frac{\mu}{2}\|x_0 - x^*\|^2\right] \le \varepsilon \quad \Rightarrow \quad \boxed{N = \tilde{\mathcal{O}}\left(\sqrt{\frac{L}{\mu}}\right).}$$

**From terms ②, ④** we find the smoothing parameter $h$:

$$\text{②}: \quad \frac{4d\kappa L^2 h^2}{\sqrt{\mu L B^2}} \le \varepsilon \quad \Rightarrow \quad h^2 \lesssim \frac{\varepsilon\sqrt{\mu}}{d}B \quad \Rightarrow \quad h \lesssim \sqrt{\frac{\varepsilon\sqrt{\mu}B}{d}};$$

$$\text{④}: \quad \frac{\kappa_\beta^2 L^2 h^{2(\beta-1)}}{\sqrt{4\mu L}} \le \varepsilon \quad \Rightarrow \quad h^{2(\beta-1)} \lesssim \varepsilon\sqrt{\mu} \quad \Rightarrow \quad \boxed{h \lesssim (\varepsilon\sqrt{\mu})^{\frac{1}{2(\beta-1)}}.}$$

**From term ③,** we find the maximum noise level $\Delta$ (via batch size $B$) at which Algorithm 1 can still achieve $\varepsilon$ accuracy:

$$\frac{\kappa d^2 \Delta^2}{h^2\sqrt{\mu L B^2}} \le \varepsilon \quad \Rightarrow \quad \Delta^2 \lesssim \frac{(\varepsilon\sqrt{\mu})^{1+\frac{1}{\beta-1}}B}{d^2} \quad \Rightarrow \quad \boxed{\Delta \lesssim \frac{(\varepsilon\sqrt{\mu})^{\frac{\beta}{2(\beta-1)}}B^{1/2}}{d}.}$$

or let's represent the batch size $B$ via the noise level $\Delta$:

$$\frac{\kappa d^2 \Delta^2}{h^2 \sqrt{\mu L B^2}} \leq \varepsilon \quad \Rightarrow \quad B \gtrsim \frac{\kappa d^2 \Delta^2}{(\varepsilon \sqrt{\mu})^{1 + \frac{1}{\beta - 1}}} \quad \Rightarrow \quad B = \mathcal{O}\left(\frac{d^2 \Delta^2}{(\varepsilon \sqrt{\mu})^{\frac{\beta}{\beta - 1}}}\right).$$

Then the oracle complexity $T = N \cdot B$ in this case has the following form:

$$\boxed{T = \max\left\{\tilde{\mathcal{O}}\left(\sqrt{\frac{d^2 L}{\mu}}\right), \tilde{\mathcal{O}}\left(\frac{d^2 \Delta^2}{(\varepsilon \mu)^{\frac{\beta}{\beta - 1}}}\right)\right\}.}$$

$\square$

