# OpenReview forum: "Maximum Noise Level as Third Optimality Criterion in Black-box Optimization Problem"
_ICLR.cc/2025/Conference — Submitted to ICLR 2025_

### Official Review · Reviewer_ogRS · 2024-10-24

**Soundness:** 3
**Presentation:** 3
**Contribution:** 3
**Rating:** 6
**Confidence:** 2

**Summary:**

This paper presents a comprehensive technique for creating a gradient-free algorithm that leverages increased function smoothness through kernel approximation, which generalizes convergence results for accelerated stochastic gradient descent to cases where the gradient oracle is biased and illustrates how bias accumulates in the algorithm's convergence.

This paper also resolves the question of iteration complexity by providing an improved estimate (as shown in Table 1) that is optimal. It determines the maximum noise level ∆ at which the algorithm can still achieve the desired accuracy ε (see Table 1 and Theorem 3.1).

Furthermore, the paper emphasizes the importance of considering the maximum noise level ∆ as a third optimality criterion alongside the standard two.

**Strengths:**

The work seems to be solid.

The paper is organized clearly.

An interesting idea on Kernel approximation is presented. A novel algorithm is given for zero-order optimization. Numerical experiments are conducted to illustrate the advantages of the proposed algorithm.

**Weaknesses:**

The definition of Oracle complexity should be introduced in the introduction, explaining its difference from Iteration complexity.

I am sorry I am not very familiar with this topic. I will change my score based on the comments of other more senior reviewers.

**Questions:**

Is the strong convexity assumption necessary? Is it possible that the proposed algorithm will perform equally well on convex or non-convex problems?

Is the bounded bias assumption, Assumption2.3, too strong?

---

> ### Author Response · Authors · 2024-11-13
> **(1/2) Response to review**
>
> Dear **Reviewer ogRS**,
>
> We appreciate the constructive feedback from the reviewer. Below are the point-to-point responses to the comments.
>
> >**The definition of Oracle complexity should be introduced in the introduction, explaining its difference from Iteration complexity.**
>
> With all due respect to you, we did not quite understand this comment because line 50 introduces Iteration complexity and line 51 introduces oracle complexity.
>
> >**Is the bounded bias assumption, Assumption2.3, too strong?**
>
> This assumption is standard and general among possible constraints. We restrict the bias to a constant. However, other types of constraints can be considered, e.g., adaptively varying with iterations proportional to the gradient norm, etc. We leave the case of other types of constraints for future work.
>
> >**Is the strong convexity assumption necessary? Is it possible that the proposed algorithm will perform equally well on convex or non-convex problems?**
>
> We do find this question quite interesting. In this paper we consider the strongly convex case, since we continue a number of previous works in this setting of the problem. Nevertheless, we are ready to add as remarks in the discussion of the theorem (if you recommend), the following results:
> 1) it is not difficult to show that the results of Theorem 3.1 generalize to the convex case, preserving the same dependence on $B$, namely in the case $B \in [1; 4d\kappa]$ and $h \lesssim \varepsilon^{3/4}$ we have the following convergence estimates for Algorithm 1:
>
> $N = \mathcal{O} \left( \sqrt{\frac{d^2 L R^2}{B^2 \varepsilon}} \right)$; $T = \mathcal{O} \left( \sqrt{\frac{d^2 L R^2}{\varepsilon}} \right)$ and $\Delta \lesssim \frac{\varepsilon^{3/2}}{\sqrt{d}}$.
>
> We can also observe that the optimal estimate of iteration complexity in the convex setup is achieved when $B=4d\kappa$. Moreover, the maximum noise level behaves in a similar way:
>
> $N = \mathcal{O}\left( \sqrt{\frac{L R^2}{\varepsilon}} \right); T = \max \left[ \mathcal{O}\left( \sqrt{\frac{d^2 L R^2}{\varepsilon}} \right), \mathcal{O} \left( \frac{d^2 \Delta^2}{\varepsilon^{2 + \frac{2}{\beta -1}}} \right) \right]$ and $\Delta \lesssim \frac{\varepsilon^{\frac{3\beta + 1}{4(\beta - 1)}}}{d} B^{1/2}$.
>
> It can be seen that if we take $\mu \sim \varepsilon$, the oracle complexity is the same in the worst case, and the maximum noise level is inferior depending on the order of smoothness compared to the strongly convex set (which is surprising).
>
> 2) We can also show that when considering deterministic adversarial noise ($|\tilde{\xi}(x)| \leq \Delta$) in a zero-order oracle instead of stochastic, Theorem 3. 1 will preserve the results except for the maximum noise level:
>
> $\Delta \lesssim \frac{(\varepsilon \sqrt{\mu})^{\frac{\beta}{2(\beta - 1)}}}{d}B^{1/2} \rightarrow  \Delta \lesssim \frac{(\varepsilon \sqrt{\mu})^{\frac{\beta}{2(\beta - 1)}}}{d}$.
>
> This can be explained by the fact that deterministic noise is more adversarial because it accumulates not only in the second moment of the gradient approximation, but also in the bias! The results in the convex case will change similarly.
>
> 3) The following may also be of interest: Given that Algorithm 1 in strongly convex setting demonstrates a linear convergence rate and employs a randomization (see e.g. $\textbf{e} \in S^d(1)$), we can derive exact estimates of high deviation probabilities using Markov’s inequality [1]:
>
> $\mathcal{P} \left( f(x_{N_{(\varepsilon \omega)}}) - f^* \geq \varepsilon  \right)  \leq \omega \frac{\mathbb{E}\left[ f(x_{N_{(\varepsilon \omega)}}) \right] - f^*}{\varepsilon \omega} \leq \omega$.

---

> ### Author Response · Authors · 2024-11-13
> **(2/2) Response to review**
>
> 4) Finally, regarding the nonconvex setup: we did not consider the nonconvex setup in our work, since we are more interested in guaranteeing global convergence. Nevertheless, we can also show that our algorithm will have global convergence for a subclass of non-convex functions that satisfy the Polyak—Lojasiewicz (PL) condition (see [2]). It is not hard to see that the results will have a similar dependence on the batch size:
>
> $N = \mathcal{\tilde{O}}\left( \frac{d}{B} \tilde{\mu}^{-1} \right)$; $T = \mathcal{\tilde{O}}\left( d \tilde{\mu}^{-1} \right)$ and $\Delta \lesssim \frac{\varepsilon \tilde{\mu}}{\sqrt{d}}$,
>
> where $\tilde{\mu}$ from PL Assumption (see [2]).
> We can also observe that the optimal estimate of iteration complexity in the convex setup is achieved when $B=4d\kappa$. Also, the maximum noise level behaves similarly:
>
> $N = \mathcal{\tilde{O}}\left( \tilde{\mu}^{-1} \right)$; $T = \max \left[ \mathcal{\tilde{O}} \left( d \tilde{\mu}^{-1} \right), \mathcal{\tilde{O}}\left( \frac{d^2 \Delta^2}{\varepsilon^{\frac{\beta}{\beta -1}} \tilde{\mu}^{\frac{2 \beta - 1}{\beta - 1}}} \right) \right]$ and $\Delta \lesssim \frac{(\varepsilon \tilde{\mu})^{\frac{\beta}{2(\beta - 1)}}}{d} B^{1/2}$.
>
> Similarly to the cases discussed above, when considering deterministic adversarial noise, the dependence on the batch size will disappear in the estimation of the maximum noise level. The transition to High probability deviations bounds is also valid. And if we compare with the estimates of Theorem 3.1, provided $\mu \sim \varepsilon$ from the strong convexity condition, and $\tilde{\mu} \sim \varepsilon$ from the PL condition, then the iteration complexity is the same, but the oracle complexity in the PL case is inferior to the strongly convex case. This can be explained by the fact that the PL condition covers a subclass of non-convex functions.
>
> *To summarize, we believe that the current results are self-sufficient, but we agree that adding these results as remarks to the discussion will improve the understanding of our results, as well as the importance of considering maximum noise as a third optimality criterion. If you recommend including these remarks in the final version of the paper, we will certainly do so!*
>
> >**I am sorry I am not very familiar with this topic. I will change my score based on the comments of other more senior reviewers.**
>
> **If you agree that we managed to address all issues, please consider raising your grade to support our work. If you believe this is not the case, please let us know so that we have a chance to respond.**
>
> [1] A Anikin et al.. Modern efficient numerical approaches to regularized regression problems in application to traffic demands matrix calculation from link loads. In Proceedings of International conference ITAS-2015., 2015.
>
> [2] Hamed Karimi, Julie Nutini, Mark Schmidt. Linear Convergence of Gradient and Proximal-Gradient Methods Under the Polyak- Lojasiewicz Condition. In Machine Learning and Knowledge Discovery in Databases: European Conference. 2016.
>
> With Respect,
>
> Authors

---

> > ### Author Response · Authors · 2024-12-03
> >
> > Dear **Reviewer ogRS**,
> >
> > The discussion period has come to an end and we still have not received any feedback from you on our responses and the revised paper. However, you can still edit your review (add text to it and change the grade).
> >
> > **Therefore, if you agree that we managed to address all issues, please consider raising your grade to support our work. If you believe this is not the case, please let us know so that we have a chance to respond.**
> >
> > With Respect,
> >
> > Authors

---

### Official Review · Reviewer_aZP7 · 2024-11-03

**Soundness:** 2
**Presentation:** 2
**Contribution:** 3
**Rating:** 5
**Confidence:** 3

**Summary:**

This paper is devoted to the study of the black-box optimization problem and the objective function value contains stochastic noise. For the case where the objective function is strongly convex and exhibits higher-order smoothness, the authors propose a novel zeroth-order accelerated batch stochastic gradient descent algorithm (ZO-ABSGD). The authors generalize existing convergence results for accelerated stochastic gradient descent to the case where the gradient oracle is biased. In addition, the authors provide improved iteration complexity through theoretical analysis and conduct a detailed examination of the maximum noise level. Finally, the authors demonstrate the performance of the proposed algorithm through experiments.

**Strengths:**

1.Compared to previous studies, the proposed algorithm improves iteration complexity and provides a thorough analysis of the maximum noise level.

2.The authors provide solid proof details to support their proposed theory.

**Weaknesses:**

1.I have some unclear aspects regarding certain writing details in this paper.

2.The experiments are insufficient; it is recommended that the authors provide additional experiments.

**Questions:**

1.The experiments demonstrate that the proposed ZO-ABSGD outperforms the ZO-VARAG algorithm on the a9a dataset, while exhibiting comparable performance to the ARDFDS algorithm. It is hoped that the authors can explain the unique practical advantages of the ZO-ABSGD algorithm in comparison to the ARDFDS algorithm.

2.I understand that the authors' main contribution lies in the theoretical aspects; however, the authors should attempt to conduct experiments on more datasets to demonstrate the efficiency of the ZO-ABSGD algorithm.

3.In section 4, the authors propose ''we show the importance of considering the maximum noise level ∆ as a third optimality criterion along with the standard two.'' It is hoped that the authors can provide a more detailed description of what "third optimality criterion" and "the standard two" specifically refer to, as well as how they are represented in the experiments.

---

> ### Author Response · Authors · 2024-11-13
> **Response to review**
>
> Dear **Reviewer aZP7**,
>
> We appreciate the constructive feedback from the reviewer. Below are the point-to-point responses to the comments.
>
> >**I have some unclear aspects regarding certain writing details in this paper.**
>
> We would be grateful if you could provide us with clarification regarding your concerns. This will allow us to better understand them.
>
> >**The experiments are insufficient; it is recommended that the authors provide additional experiments.**
>
> Although our work is theoretical, as is customary in the optimization community, we have added experiments that are of interest not only to the optimization community but also to the machine learning community in general. Moreover, we have included these experiments to confirm our theoretical results, in particular, that maximum noise level is worth considering as a third optimality criterion on par with iteration and oracle complexity. Regarding additional experiments, it seems to us that there is no need to add anything else, but if you suggest a specific problem statement for the experiments (model or close to reality), we are ready to consider adding additional experiments.
>
> >**The experiments demonstrate that the proposed ZO-ABSGD outperforms the ZO-VARAG algorithm on the a9a dataset, while exhibiting comparable performance to the ARDFDS algorithm. It is hoped that the authors can explain the unique practical advantages of the ZO-ABSGD algorithm in comparison to the ARDFDS algorithm.**
>
> This is indeed an important question, as Figure 3 demonstrates both the importance of considering the maximum noise level as a third optimality criterion, as well as the advantages of using the kernel approximation. The answer to this question can be found in the discussion of Figure 3 (see line 463 and following), but we will respectfully provide a brief clarification: the behavior of ZO-VARAG can be explained by the fact that the algorithm is not tuned for the case where the zero-order oracle is subject to adversarial noise, which is why it converges to the error floor so quickly. Regarding ARDFDS, this algorithm is tuned to more general noise, namely deterministic noise, which is more adversarial because this noise accumulates not only in the second moment of the gradient approximation, but also in the bias. Therefore, the superiority of our algorithm can be explained by the fact that our algorithm makes use of increased smoothness information (via kernel approximation), in particular $\beta = 3$ , while ARDFDS uses $L_2$ randomization, which only works in the smoothness condition $\beta = 2$ and does not make any use of increased smoothness information.
>
> >**I understand that the authors' main contribution lies in the theoretical aspects; however, the authors should attempt to conduct experiments on more datasets to demonstrate the efficiency of the ZO-ABSGD algorithm.**
>
> We have answered this question above.
>
> >**In section 4, the authors propose ''we show the importance of considering the maximum noise level $\Delta$ as a third optimality criterion along with the standard two.'' It is hoped that the authors can provide a more detailed description of what "third optimality criterion" and "the standard two" specifically refer to, as well as how they are represented in the experiments.**
>
> The main goal of our work is to show the importance of considering the maximum noise level in gradient-free algorithms. Throughout the first two sections, we provided motivation in finding the maximum noise level. In Section 3, we demonstrated a surprising way to improve the estimate on the noise level, thereby showing that the maximum noise level can vary with the order of smoothness (this is not obvious!). And already in Section 4, we validated our theoretical results on a numerical experiment, in particular the importance of creating and analyzing an optimization algorithm that is robust to adversarial noise, as demonstrated by the ZO-VARAG algorithm. As we see with ARDFDS and ZO-ABSGD, the smoothness order does affect the maximum noise level, in particular, since we fixed the noise level, the higher the smoothness order, the better the error floor, as we can see in Figure 3.
>
>
> **If you agree that we managed to address all issues, please consider raising your grade to support our work. If you believe this is not the case, please let us know so that we have a chance to respond.**
>
>
> With Respect,
>
> Authors

---

> > ### Author Response · Authors · 2024-12-03
> >
> > Dear **Reviewer aZP7**,
> >
> > The discussion period has come to an end and we still have not received any feedback from you on our responses and the revised paper. However, you can still edit your review (add text to it and change the grade).
> >
> > **Therefore, if you agree that we managed to address all issues, please consider raising your grade to support our work. If you believe this is not the case, please let us know so that we have a chance to respond.**
> >
> > With Respect,
> >
> > Authors

---

### Official Review · Reviewer_METE · 2024-11-05

**Soundness:** 3
**Presentation:** 3
**Contribution:** 3
**Rating:** 5
**Confidence:** 4

**Summary:**

This paper generalizes the analysis of an accelerated SGD algorithm in Vaswani et al. (2019) to allow for biased gradient oracle ands mini-batch data. Base on this contribution, this paper develops a zeroth-order method based on Kernel approximation and prove new convergence results. Experiments show that this new method is effective.

**Strengths:**

1. This paper generalizes an existing accelerated SGD method in Vaswani et al. (2019) to the case with bised gradient noise.

**Weaknesses:**

1. Biased noise has been studied in existing works such as Akhavan, Chzhen, Pontil, Tsybakov (2023), see their assumption B. Thus, this is not new at least for zeroth order method.
2. The convergence rate is achieved by taking $\Delta$ linear in target accuracy $\epsilon$, which makes it hard to compare with other listed works which treat $\Delta$ as some $\epsilon$-independent problem-dependent constant.
3. The writing of this paper is not smooth, some terms come up without definition.

**Questions:**

1. When mentioning Kernel approximation in the introduction, can you briefly discuss the advantage of using this approximation?
2. In definition 1.3, $e$ and $r$ are undefined. Do you mean there are two queries at the same point $x_k$?
3. The lemma 2.4 and lemma 2.5 presented results for an algorithm not described? And what is $\eta, \tilde{R}$, why the third with $\tilde{R}$ does not affect much? Can you revise and text and make these clear?
4. Can the authors briefly discuss what new tricks used in this paper to allow biased noise oracles?

---

> ### Author Response · Authors · 2024-11-13
> **(1/2) Response to review**
>
> Dear **Reviewer METE**,
>
> We appreciate the constructive feedback from the reviewer. Below are the point-to-point responses to the comments.
>
> >**Biased noise has been studied in existing works such as Akhavan, Chzhen, Pontil, Tsybakov (2023), see their assumption B. Thus, this is not new at least for zeroth order method.**
>
> Could you please tell us what you mean by biased noise? Our work proposes a detailed scheme to create a gradient-free algorithm based on a first-order algorithm. Since the gradient approximation, both $L_2$ randomization and kernel approximation have bias, it is important to use a first-order method that uses a biased gradient oracle as a base. In [1], an accelerated algorithm that uses an unbiased gradient oracle was proposed, which is why one of our results is a generalization of the first-order algorithm from [1] to the case with a biased gradient oracle. Next, we replace the gradient oracle with a kernel approximation to create a new gradient-free algorithm. And already in the kernel approximation we use a zero-order oracle, which is subject to stochastic adversarial noise. And it is this concept of noise that is considered in [2]. However, both in [2] and in previous works (see Table 1), the authors “fight” for only one optimality criterion - oracle calls. In our turn, we consider all three optimality criteria: we achieve optimal iteration complexity, we do not worsen oracle complexity, and, more importantly, we explicitly estimate the maximum noise level $\Delta$. In previous works, the authors did not provide any estimate on the allowable noise level. While our work tells us that it is important to consider the maximum noise level as a third optimality criterion, on par with iteration and oracle complexity!
>
> >**The convergence rate is achieved by taking $\Delta$ linear in target accuracy $\epsilon$, which makes it hard to compare with other listed works which treat $\Delta$ as some $\epsilon$-independent problem-dependent constant.**
>
> Yes, previous work has not addressed in any way the question of how much noise can be added to a zero-order oracle so that the algorithm still converges well. This is probably due to the fact that this concept of noise accumulates only in the second moment and the error floor will improve as the batch size increases (see Theorem 3.1 for the case when $B>4d\kappa$). However, if we consider another concept of noise, namely deterministic noise ($|\tilde{\xi}(x)| \leq \Delta$), then such noise will accumulate not only in the second moment but also in the bias of the gradient approximation, thus the estimate for the maximum noise level in the case $B > 4d\kappa$ will not depend on the batch size: $\Delta \lesssim \frac{(\varepsilon \sqrt{\mu})^{\frac{\beta}{2(\beta - 1)}}}{d}$. Hence, what we have been able to show is that the maximum noise level in the case $B> 4 d \kappa$ can improve with increasing smoothness order is not an obvious result. That is why we describe the importance of finding the maximum noise level.
>
> >**The writing of this paper is not smooth, some terms come up without definition.**
>
> We would appreciate it if you could point out these typos.
>
> >**When mentioning Kernel approximation in the introduction, can you briefly discuss the advantage of using this approximation?**
>
> Perhaps the main advantage of the kernel approximation is that the gradient approximation can utilize higher smoothness information by requiring information at only two points. For example, in [3], the authors use a higher-order finite-difference approximation, which requires more calls to the (gradient-free) oracle at each iteration, whereas while our approach uses kernel estimation, requiring only two computations of the function value (realization) per iteration.
>
> >**In definition 1.3, $e$ and $r$ are undefined. Do you mean there are two queries at the same point $x_k$?**
>
> We thank the Reviewer, of course, we will specify that $\textbf{e} \in S^d(1)$ is a random vector uniformly distributed on the Euclidean unit sphere, and $r$ is a random variable uniformly distributed on the interval $[-1;1]$.

---

> > ### Author Response · Authors · 2024-11-13
> > **(2/2) Response to review**
> >
> > >**The lemma 2.4 and lemma 2.5 presented results for an algorithm not described? And what is \eta, $\tilde{R}$, why the third with $\tilde{R}$ does not affect much? Can you revise and text and make these clear?**
> >
> > We believe that the main algorithm of this paper is a gradient-free algorithm, so we have not added a scheme for the first-order algorithm. However, if you recommend, we will add the scheme for updating the first-order algorithm (where $\eta$ is the step size) for which Lemmas 2.5 and Theorem 2.6 are given. Regarding $\tilde{R}$, we will add a clarification in the Notation section. Regarding the third summand, the noise does not accumulate due to the decreasing sequence, thus not affecting convergence.
> >
> > >**Can the authors briefly discuss what new tricks used in this paper to allow biased noise oracles?**
> >
> > We think we have answered that question above.
> >
> > **If you agree that we managed to address all issues, please consider raising your grade to support our work. If you believe this is not the case, please let us know so that we have a chance to respond.**
> >
> > [1] Vaswani, S., Bach, F., & Schmidt, M.. Fast and faster convergence of sgd for over-parameterized models and an accelerated perceptron. In The 22nd international conference on artificial intelligence and statistics (pp. 1195-1204). PMLR. 2019.
> >
> > [2] Arya Akhavan, Evgenii Chzhen, Massimiliano Pontil, and Alexandre B Tsybakov. Gradient-free optimization of highly smooth functions: improved analysis and a new algorithm. arXiv preprint arXiv:2306.02159, 2023.
> >
> > [3] A. S. Berahas, L. Cao, K. Choromanski, and K. Scheinberg. A theoretical and empirical comparison of gradient approximations in derivative-free optimization. Foundations of Computational Mathematics. 2022.
> >
> > With Respect,
> >
> > Authors

---

> ### Comment · Reviewer_METE · 2024-11-22
>
> I thank the authors for their responses. I have provided specific examples highlighting why the writing of this paper needs improvement to help readers follow the exposition, particularly for Lemma 2.4 and Lemma 2.5. Furthermore, I remain unconvinced by the authors’ arguments regarding the quantification of the maximum allowable noise. If the noise is linear in $\epsilon$, it seems to me that the noise should approach zero if we aim to truly solve the optimization problem. As such, I do not see the contribution in this aspect.

---

> ### Author Response · Authors · 2024-11-23
> **Official Comment to the Reviewer METE**
>
> Dear **Reviewer METE**,
>
> We thank you for your prompt response!
>
> >**I have provided specific examples highlighting why the writing of this paper needs improvement to help readers follow the exposition, particularly for Lemma 2.4 and Lemma 2.5.**
>
> Regarding Lemma 2.4 and Lemma 2.5, we would like to draw your attention that Lemma 2.4 is not in our paper, we have Assumption 2.4 (noise boundedness), Lemma 2.5 (results of [1]) and Theorem 2.6 (generalization of the results of [1] to the case with a biased gradient oracle and adding a batching technique). As we answered in the previous message, we will add to the final version of the paper information on both the update scheme of the original algorithm [1] and $\eta, \tilde{R}$, as well as why the third summand does not affect convergence (see the previous message for details).
>
> >**Furthermore, I remain unconvinced by the authors’ arguments regarding the quantification of the maximum allowable noise. If the noise is linear in $\epsilon$, it seems to me that the noise should approach zero if we aim to truly solve the optimization problem. As such, I do not see the contribution in this aspect.**
>
> With all due respect, we would like to clarify a point regarding the results of Theorem 3.1. Indeed, in the current concept of noise (given that $\Delta > 0$) we can achieve any desired accuracy, provided that we use any large batch size (which is demonstrated in the maximum noise estimate: $\Delta \leq \frac{(\varepsilon \sqrt{\mu})^{\frac{\beta}{2(\beta -1 )}}}{d} B^{1/2}$. However, as is often the case in life, we can't use any batch size we want, because the batch size directly affects the number of oracle calls (which is often limited). If we have a finite batch size, then as Theorem 3.1 shows, our algorithm will already converge only to error floor (it will not converge to any desired accuracy). We would like to point out that our result for maximum noise shows that the accuracy of convergence (error floor) can be improved through the order of smoothness of the function, i.e. the higher the smoothness of the function, the more accurately the algorithm will converge. This result is not trivial!
>
> **If you agree that we managed to address all issues, please consider raising your grade to support our work. If you believe this is not the case, please let us know so that we have a chance to respond.**
>
> [1] Vaswani, S., Bach, F., & Schmidt, M.. Fast and faster convergence of sgd for over-parameterized models and an accelerated perceptron. In The 22nd international conference on artificial intelligence and statistics (pp. 1195-1204). PMLR. 2019.
>
> With Respect,
>
> Authors

---

> > ### Author Response · Authors · 2024-12-03
> >
> > Dear **Reviewer METE**,
> >
> > The discussion period has come to an end and we still have not received any feedback from you on our responses and the revised paper. However, you can still edit your review (add text to it and change the grade).
> >
> > **Therefore, if you agree that we managed to address all issues, please consider raising your grade to support our work. If you believe this is not the case, please let us know so that we have a chance to respond.**
> >
> > With Respect,
> >
> > Authors

---

### Official Review · Reviewer_nr6M · 2024-11-06

**Soundness:** 2
**Presentation:** 1
**Contribution:** 1
**Rating:** 3
**Confidence:** 3

**Summary:**

The paper analyzes the zeroth order method with a biased oracle for strongly convex functions with higher-order smoothness. The author introduces ZO Accelerated batched SGD and demonstrates that it achieves the optimal iteration complexity. The paper highlights that its results provide estimates of  the maximum noise level required for the algorithm to converge.

**Strengths:**

The paper extends the analysis of the first-order Accelerated SGD to a setting with a biased gradient oracle and the use of mini-batches. Further, It combines this result with the estimated gradients from zeroth-order orcale.

**Weaknesses:**

1. The paper’s writing is problematic. Symbols are introduced before their definitions, such as $e$ and $r$ in Definition 1.3. The first-order Accelerated SGD algorithm is presented without a definition. Furthermore, Figure 1 seems unnecessary for a theoretical work in this field.

2. Technically, the contribution is limited. The convergence of Accelerated SGD has been proven, and extending it to the biased and mini-batch settings appears straightforward. Additionally, the estimations of bias and second moment of the zeroth-order oracle are well-established in the literature.

3. The motivation behind considering the maximum noise level is unclear. Previous work assumed a constant $\Delta$ and showed convergence to minima. This work considers $\Delta$ diminishing with the target accuracy, and does not converge to minima with constant $\Delta$. I fail to see the advantage of considering this case. Furthermore, Table 1 does not provide a fair comparison since $\Delta$ depends on $\epsilon$ in this work. Moreover, Table 1 reports iteration complexity, while the proposed algorithm utilizes mini-batches, which also leads to an unfair comparison. The oracle complexity should be the more relevant measure in this context.

**Questions:**

1. Could the author clarify the technical challenges in this work and address the third point in Weakness?

2. Regarding the Zeroth order oracle, how can we guarantee that $\xi_1 \neq \xi_2$? If we know their values, we can already get the true function value.

3. In Figure 2 b), why does the red curve converge to minimizers even in the case of bias?

4. In the numerical experiment section, what noise is used? Why is the final accuracy of ZO-VARAG large, given that the noise level is already set to be small?

---

> ### Author Response · Authors · 2024-11-13
> **(1/3) Response to review**
>
> Dear **Reviewer nr6M**,
>
> Thank you for your feedback. Please find our responses below.
>
> >**The paper’s writing is problematic. Symbols are introduced before their definitions, such as $e$ and $r$ in Definition 1.3. The first-order Accelerated SGD algorithm is presented without a definition. Furthermore, Figure 1 seems unnecessary for a theoretical work in this field.**
>
> We thank the Reviewer, of course, we will clarify that $\textbf{e} \in S^d(1)$ is a random vector uniformly distributed on the Euclidean unit sphere, and $r$ is a random value uniformly distributed on the interval $[-1;1]$. Regarding Figure 1, it seems to us that visually it is easier to understand the motivation for finding the maximum noise level.
>
> >**Technically, the contribution is limited. The convergence of Accelerated SGD has been proven, and extending it to the biased and mini-batch settings appears straightforward. Additionally, the estimations of bias and second moment of the zeroth-order oracle are well-established in the literature.**
>
> We position our paper as a narrative that it is worth considering the maximum noise level along with other optimality criteria of a gradient-free algorithm. The proposed problem statement has been chosen as an example on which we demonstrate the importance of finding the maximal noise level. In particular, in the first two sections, we provide motivation for finding the maximal noise level. In Section 3, we provide theoretical results, and show how the maximal noise level can be improved. And in Section 4, we prove the importance of considering the maximum noise level as the third optimality criterion in a numerical experiment. Despite the fact that we chose this problem formulation as an example (as discussed above), we were able to achieve very interesting and novel results. As correctly pointed out by the Reviewer, we generalized the results of the accelerated first-order algorithm [1] to the case with a biased gradient oracle. This generalization may be of independent interest. Moreover, we closed the question of the optimality of the iteration complexity in this formulation of the problem. Finally, we have shown that by using a large batched size one can improve the estimate on the maximum noise level, namely, that as the smoothness order of the objective function increases, the estimate of the maximum noise level improves (we believe this is a rather surprising result!). Finally, with this paper we have explicitly demonstrated a technique for creating a gradient-free algorithm.
>
> >**The motivation behind considering the maximum noise level is unclear. Previous work assumed a constant $\Delta$ and showed convergence to minima. This work considers $\Delta$ diminishing with the target accuracy, and does not converge to minima with constant $\Delta$. I fail to see the advantage of considering this case. Furthermore, Table 1 does not provide a fair comparison since $\Delta$ depends on $\epsilon$ in this work. Moreover, Table 1 reports iteration complexity, while the proposed algorithm utilizes mini-batches, which also leads to an unfair comparison. The oracle complexity should be the more relevant measure in this context.**
>
> Regarding the table, if you recommend adding a column with oracle calls, we will certainly do it. However, both in [2] and in previous works (see Table 1), the authors “fight” for only one optimality criterion - oracle calls (which are also iteration complexity). For our part, we consider all three optimality criteria: we achieve optimal iteration complexity (using the already accelerated first-order batched algorithm as a base), we do not deteriorate oracle complexity, and, more importantly, we explicitly estimate the maximum noise level $\Delta$.
>
> Nevertheless, you are right to note that previous works have not addressed in any way the question of how much noise the zero-order oracle can be noisy so that the algorithm still converges well. This is probably because this concept of noise accumulates only at the second moment and the error floor will improve as the batch size increases (see Theorem 3.1 for the case when $B>4d\kappa$). However, if we consider another concept of noise, namely deterministic noise ($|\tilde{\xi}(x)| \leq \Delta$), then such noise will accumulate not only in the second moment but also in the bias of the gradient approximation, thus the estimate for the maximum noise floor in the case $B > 4d\kappa$ will not depend on the batch size: $\Delta \lesssim \frac{(\varepsilon \sqrt{\mu})^{\frac{\beta}{2(\beta - 1)}}}{d}$. That is, the algorithm will not converge as accurately as possible, no matter how large the size of the batches. Hence, what we managed to show is that the maximum noise level in the case $B> 4 d \kappa$ can improve with increasing smoothness order is not an obvious result. That is why we describe the importance of finding the maximum noise level.

---

> > ### Author Response · Authors · 2024-11-13
> > **(2/3) Response to review**
> >
> > >**Could the author clarify the technical challenges in this work and address the third point in Weakness?**
> >
> > Regarding the third point, we answered above. Regarding challenges, as we described in our paper that in this problem formulation we were motivated by two questions, whether an optimal estimate of iterative complexity can be achieved and how much can we noise the zero-order oracle so that our algorithm still converges well. In the process of addressing these questions, we faced the difficulty of choosing a first-order algorithm on which to base our algorithm. Although the original problem is deterministic, we need to choose a first-order algorithm that uses a biased stochastic gradient oracle, since our gradient approximation contains artificial stochasticity as well as bias (see (5)). Hence, this is how Theorem 2.6 came about. But perhaps what can really be called a challange is this: From Section 2 (in particular Theorem 2.6), we see that the algorithms converge to an error floor (which can be controlled in a gradient-free setup via maximum noise level) due to the accumulation of inexactness. In this paper, we consider an optimization problem where the function is not just smooth, but has increased smoothness. We use kernel approximation to account for the advantage of the increased smoothness information. In Theorem 3.1, we show how we can improve the iteration complexity in the strongly convex formulation, and that we can achieve an optimal estimate by batching ($B=4dκ$, the iteration number estimate is optimal). **A further increase of the batch size seems unreasonable**, since the iteration complexity $N$ is no longer improveable, but the oracle complexity $T$ will only worsen... However, in our work we consider yet the maximum noise level $\Delta$, which in the case of a full batch, as we see, does not depend on the smoothness order, i.e., when using the proposed algorithm (in the case, for example, when the smoothness order of the objective function $\beta = 5$ and the batch size $B \in [1, 4d\kappa]$) we will converge to the **same error floor** as existing algorithms (solving smooth problems $\beta=2$), only faster. However, we want our algorithm to not only converge faster, but also more accurately (to achieve a better error floor). **This is exactly what we have achieved** (which is actually non-trivial). We show that when $B > 4d\kappa$ is used, the maximum noise level (or at a fixed noise level, error floor) accounts for smoothness order, while maintaining optimality in the iteration complexity.
> >
> > >**Regarding the Zeroth order oracle, how can we guarantee that $\xi_1 \neq \xi_2$? If we know their values, we can already get the true function value.**
> >
> >
> > We consider a zero-order oracle as a black box, i.e., when accessed, the oracle produces only one value, which is the noisy objective function. An example of such an oracle can be a human, i.e., $\tilde{f}_p$ is the $p$-th person of the group, then we can confidently guarantee that $\xi_1 \neq \xi_2$. Such an example can be tied, for example, to the currently popular application: RLHF.

---

> > > ### Author Response · Authors · 2024-11-13
> > > **(3/3) Response to review**
> > >
> > > >**In Figure 2 b), why does the red curve converge to minimizers even in the case of bias?**
> > >
> > > As mentioned on line 288 and below, the red line characterizes the convergence of the algorithm using the unbiased gradient oracle, and the blue line characterizes the convergence of the first order algorithm using the biased gradient oracle. Figure 2b presents the two cases so that we can see the difference between the cases, namely that the convergence rates are the same, but the accuracy to which these algorithms converge is different.
> > >
> > > >**In the numerical experiment section, what noise is used? Why is the final accuracy of ZO-VARAG large, given that the noise level is already set to be small?**
> > >
> > > The answer to this question can be found in the discussion of figure 3 (see line 463ff.), but we will respectfully provide a brief clarification: the behavior of ZO-VARAG can be explained by the fact that the algorithm is not tuned to the case where the zero-order oracle is subject to adversarial noise, which is why it converges to the error floor so quickly. Regarding ARDFDS, this algorithm is tuned to more general noise, namely deterministic noise, which is more adversarial because this noise accumulates not only in the second moment of the gradient approximation, but also in the bias. Therefore, the superiority of our algorithm can be explained by the fact that our algorithm makes use of increased smoothness information (via kernel approximation), in particular $\beta = 3$, while ARDFDS uses $L_2$ randomization, which only works in the smoothness condition $\beta = 2$ and does not make any use of increased smoothness information.
> > >
> > > Moreover, even if we were to consider the case where $\Delta = 0$, the adversarial noise would be machine accuracy (mantissa).
> > >
> > > **If you agree that we managed to address all issues, please consider raising your grade to support our work. If you believe this is not the case, please let us know so that we have a chance to respond.**
> > >
> > >
> > > [1] Vaswani, S., Bach, F., & Schmidt, M.. Fast and faster convergence of sgd for over-parameterized models and an accelerated perceptron. In The 22nd international conference on artificial intelligence and statistics (pp. 1195-1204). PMLR. 2019.
> > >
> > >
> > > [2] Arya Akhavan, Evgenii Chzhen, Massimiliano Pontil, and Alexandre B Tsybakov. Gradient-free optimization of highly smooth functions: improved analysis and a new algorithm. arXiv preprint arXiv:2306.02159, 2023.
> > >
> > > With Respect,
> > >
> > > Authors

---

> > > > ### Author Response · Authors · 2024-12-03
> > > >
> > > > Dear **Reviewer nr6M**,
> > > >
> > > > The discussion period has come to an end and we still have not received any feedback from you on our responses and the revised paper. However, you can still edit your review (add text to it and change the grade).
> > > >
> > > > **Therefore, if you agree that we managed to address all issues, please consider raising your grade to support our work. If you believe this is not the case, please let us know so that we have a chance to respond.**
> > > >
> > > > With Respect,
> > > >
> > > > Authors

---

### Author Response · Authors · 2024-11-15
**Reply to all reviewers**

We would like to thank the reviewers for their valuable comments that help to improve our manuscript. We addressed comments from each of the reviewers separately below each of the reviews.

With Respect,

Authors

---

> ### Author Response · Authors · 2024-11-28
>
> Dear **Reviewers**,
>
> We have taken into account all the comments and made the minor changes we promised to make to our article.
>
> **If you agree that we managed to address all issues, please consider raising your grade to support our work. If you believe this is not the case, please let us know so that we have a chance to respond.**
>
> With Respect,
>
> Authors

---

### Meta-Review · Area_Chair_Mcp5 · 2024-12-18

**Metareview:**

The paper studies black box zeroth order optimization with a stochastic zeroth-order oracle. It generalizes a first-order algorithm (an accelerated SGD) from prior work to handle biased stochastic gradient oracles. It provides new complexity results, attaining optimal iteration and oracle complexities and quantifying the maximum level of noise that can be supported. While the paper does have some interesting contributions and may merit publication at one of the top conferences, in its current form it is not written sufficiently well and it was hard to judge its novelty. As a result, there was not enough support to push it towards acceptance.

**Additional Comments On Reviewer Discussion:**

The authors tried to address the reviewers' points, however the response was insufficient to convince any of the reviewers that the paper is above the acceptance threshold.

---

### Decision · Program_Chairs · 2025-01-22

Reject